# A particle-field approach bridges phase separation and collective motion in active matter

Robert Großmann [1,2], Igor S. Aranson [3 ✉] & Fernando Peruani [1,4 ✉]

Whereas self-propelled hard discs undergo motility-induced phase separation, self-propelled rods exhibit a variety of nonequilibrium phenomena, including clustering, collective motion, and spatio-temporal chaos. In this work, we present a theoretical framework representing active particles by continuum fields. This concept combines the simplicity of alignment-based models, enabling analytical studies, and realistic models that incorporate the shape of self-propelled objects explicitly. By varying particle shape from circular to ellipsoidal, we show how nonequilibrium stresses acting among self-propelled rods destabilize motility-induced phase separation and facilitate orientational ordering, thereby connecting the realms of scalar and vectorial active matter. Though the interaction potential is strictly apolar, both, polar and nematic order may emerge and even coexist. Accordingly, the symmetry of ordered states is a dynamical property in active matter. The presented framework may represent various systems including bacterial colonies, cytoskeletal extracts, or shaken granular media.

[1] Laboratoire J.A. Dieudonné, Université Côte d'Azur, UMR 7351 CNRS, 06108 Nice, France. [2] Institute of Physics and Astronomy, University of Potsdam, D-14476 Potsdam, Germany. [3] Department of Biomedical Engineering, Pennsylvania State University, University Park, PA 16802, USA. [4] Laboratoire de Physique Théorique et Modélisation, UMR 8089, CY Cergy Paris Université, 95302 Cergy-Pontoise, France. ✉email: isa12@psu.edu; fernando.peruani@cyu.fr

Interacting self-propelled particles, termed active matter, are the standard model of collective behavior out of thermo-dynamic equilibrium[1]. Active systems become increasingly popular in different disciplines studying diverse systems from phoretic colloids[2,3], self-organized collective swimming of bacteria[4], and self-assembly in biomimetic systems[5] to collective animal behavior[6,7].

The current theoretical understanding of active matter relies on two cornerstones. One of these is the emergence of phase-separated states in ensembles of self-propelled hard discs[8–10] due to the combined effect of self-propulsion and isotropic repulsion[11,12]. This phenomenon, sharing similarities with vibrated granular media[13], was termed motility-induced phase separation (MIPS). Its theoretical appeal stems from the potential mapping of the none-quilibrium dynamics at large scales to an effective equilibrium theory for the density field[14–17]. Despite various realizations of self-propelled discs were designed[2,18], experimental evidence of this very type of active phase separation is still lacking. Furthermore, recent experiments with active Janus colloids[19] support the hypothesis that torques leading to orientational ordering within clusters interrupt MIPS. Note that torques among self-propelled discs are only neg-ligible if tangential friction and dipole–dipole (electric or magnetic) interactions are absent. Moreover, long-ranged hydrodynamic flows can mediate nontrivial orientational interactions[20] that substantially depend on the geometry of the system and its boundary conditions; for an in-depth analysis how hydrodynamic interactions suppress MIPS, see ref. [21] and references therein.

Orientational symmetry breaking and the emergence of collective motion due to velocity alignment is another cornerstone of active matter[22–27]. Effective alignment interactions may occur, for example, due to inelastic collisions of colloidal particles[28] observed in ensemble of vertically shaken discs[18], due to hydrodynamic interactions[20,21], or by combined interactions such as hydrodynamic-electric ones in Quicke rollers[29] and hydrodynamic-magnetic ones in magnetic rollers[30,31]. However, the most elemental and ubiquitous source of alignment is given by the particle shape, i.e., by anisotropic repulsion among spatially extended, self-propelled objects[27,32]. Recent numer-ical studies unveiled a large variety of collective phenomena among self-propelled, rod-shaped particles including mesoscale-turbulence[33], formation of bands and aggregates[34], accumulation at confining walls[35], and a complex phase diagram depending on rigidity and density[36,37].

Beyond the inherent theoretical interest in the physics of self-propelled rods, there exists a large number of applications: motile bacteria in two-dimensions[4,38–40], biomimetic systems like motility assays[41–43] as well as shaken granular rods[44–46]. Note in this context that the collective dynamics of cells[47], modeled by soft deformable active particles[48,49] or phase fields[50], constitutes a physically different class of systems due to the inherent coupling of particle shape and the level of activity—for vanishing activity, cells become circular and, thus, these models do not behave like a system of passive rods in this limit; particularly, they do not exhibit an isotropic-nematic transition expected to emerge as a result of free energy minimization in passive liquid crystals[51].

In short, particle shape controls the physics of active systems: whereas active phase separation is expected for self-propelled discs[12], the coupling of self-propulsion and anisotropic volume exclusion leads to collectively moving clusters[27,32]. A framework encompassing all of these central phenomena can be considered the basis for a theory of active matter. In this context, novel theoretical concepts are called for as the application of tools from equilibrium statistical mechanics to active matter, e.g., pressure[52], is limited to special cases.

The complexity of models for spatially extended, anisotropic objects has hindered analytical studies and systematic coarse-graining addressing their collective properties—the characterization

has been mainly carried out by numerical simulations[27,32–37,53]. The derivation of hydrodynamic equations from microscopic models has only been possible for particles with a prescribed velocity-alignment rule[54–59], based on symmetry considerations, or heuristic interactions including Onsarger's interaction argument for rods in the limit of infinite aspect ratio[60]. Only recently, the collective dynamics of self-propelled rods has been character-ized by effective transport properties (collective speed and rotational diffusion), which has allowed to assess numerically the importance of motility-induced phase separation in those systems[61].

In this work, we present a modeling approach for self-driven objects that brings the simplicity of alignment-based models and enables analytical analysis. It provides a physically coherent, realistic modelling of self-propelled objects with steric, repulsive interactions: each individual entity is represented by an aniso-tropic, smooth field whose mutual interactions are derived from the minimization of overlap energy. Thus, the interaction force and torque result from a single interaction principle, namely the minimization of potential energy that depends explicitly on particle shape. This approach yields a universal, simple, and descriptive model that links different phenomena at the heart of active matter. Within the same framework, it consistently reproduces MIPS for self-propelled discs[12] and predicts the emergence of orientational order as well as polar clustering for self-propelled hard rods[27,32]. Thereby, it contributes to the con-nection of the realms of scalar and vectorial active matter[37,61,62]. Combining numerical simulations and analytically derived coarse-grained order parameter equations, we show how MIPS breaks down for anisotropic objects due to the combined action of self-propulsion and anisotropic repulsion. Therefore, it underpins the specific interactions that are responsible for the restabilization of homogeneous states for rod-shaped particles. The resulting nonequilibrium stresses acting on the microscale induce orientational alignment of different symmetries locally. In this system, the rigidity of particles determines the symmetry of ordered states: long-lived, giant moving clusters are observed if particles strongly repel each other to prevent particle crossing, whereas large-scale nematic order emerges for particles which can slide past each other. We further report that those regimes are separated by a bistable coexistence region, similar to the recently reported ones in motility assays[43]. Thereby, we shed light on the importance of anisotropic repulsion as a source of orientational alignment, particularly on how the interrelation of particle shape, rigidity, and self-propulsion determines emergent collective behavior—key elements to be considered in the design of bio-mimetic materials. Unifying seemingly different phenomena at the heart of active matter within one theoretical framework is expected to pave the way toward a comprehensive understanding of soft and deformable active matter such as bacterial colonies[4,63] or driven filaments[41–43].

## Results

**Field representation of active particles.** We represent individual active particles by Gaussian fields with the principle axes $l_\parallel$ and $l_\perp$ as shown in Fig. 1, reminiscent of smoothed-particle hydro-dynamics[64] and Gaussian model potentials to describe molecular interactions[65]. The main control parameter for the particle shape, which in turn determines the collective properties of this active system, is the anisotropy

$$\varepsilon = \frac{l_\parallel^2 - l_\perp^2}{l_\parallel^2 + l_\perp^2} = \frac{\phi^2 - 1}{\phi^2 + 1}. \tag{1}$$

The parameter $\varepsilon$ can equivalently be expressed by the aspect ratio $\phi = l_\parallel/l_\perp$. The basic idea of the particle-field representation

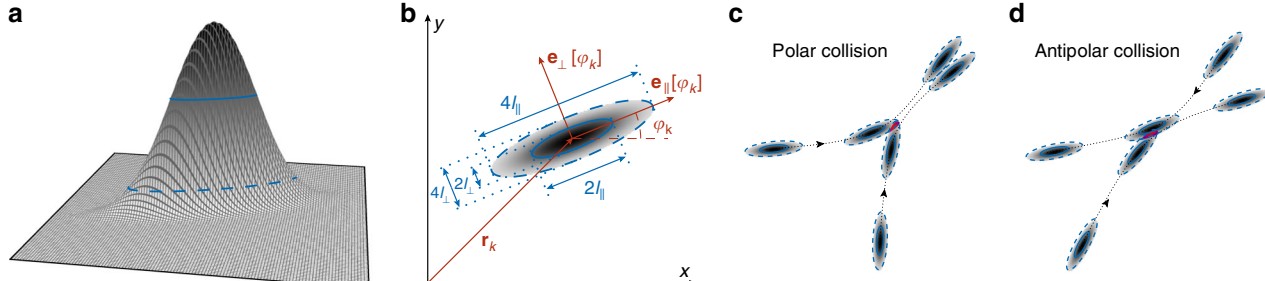

**Fig. 1 Graphical illustration of the smooth-particle approach.** The field representing one particle is shown as a three-dimensional illustration in panel (**a**) and as a top view in panel (**b**), together with the particle extensions $l_\parallel$, $l_\perp$, its orientations $\mathbf{e}_{\parallel,\perp}[\varphi_k]$ and the center of mass position $\mathbf{r}_k$. Particles are circular for $\varepsilon = 0$ ($l_\parallel/l_\perp = 1$) and become needle-shaped in the limit $\varepsilon \to 1$ ($l_\parallel/l_\perp \to \infty$). Here, $l_\parallel/l_\perp = 4$, $\varepsilon = 15/17$. Furthermore, a polar (acute) (panel **c**) and an antipolar (obtuse) collision (panel **d**) of rigid rods is illustrated. The overlap upon collision is highlighted in color. After a polar collision, rods tend to move in parallel such that their positions and orientations are highly correlated. In contrast, particles are quickly separated far from another after an antipolar collision. Therefore, the probability to find two rods moving in parallel is enhanced; in short, polar collisions are the precursor for clustering. For additional modeling details see Supplementary Note 1 and Supplementary Fig. 1. Supplementary Movies 5–8 illustrate various outcomes of binary interactions.

is that particles repel each other to minimize their mutual overlap upon encounter[32,65]. The Gaussian representation is advantageous because the overlap of two particles can be calculated analytically. In this way, we construct a repulsive binary interaction energy that is a function of the overlap; for details on the derivation of this energy and its explicit form, see Methods. The interaction force $\mathbf{f}_2$ and torque $m_2$ result from the minimization of this interaction energy. The force

$$\mathbf{f}_2(\Delta\mathbf{r}, \varphi, \varphi') = \mathcal{M}_\varepsilon(\Delta\mathbf{r}, \varphi, \varphi') \cdot \Delta\mathbf{r}, \quad (2)$$

between two rods is basically given by the relative position $\Delta\mathbf{r}$ of their centers of mass, however, it is anisotropic in view of their relative orientations, represented by the matrix $\mathcal{M}_\varepsilon$. Further, the torque consists of two contributions

$$m_2(\Delta\mathbf{r}, \varphi, \varphi') = a\sin[2(\varphi - \alpha)] + b\sin[2(\varphi' - \varphi)], \quad (3)$$

where $\alpha = \arg(\Delta\mathbf{r})$, and the prefactors $a$ and $b$ abbreviate generalized interaction strengths; for symmetry considerations, one can think of them as constants. The term with prefactor $b$ is the well-known nematic alignment of the body axes, proposed on phenomenological grounds[24,66] and analyzed within kinetic theories[57,59,67]. The term proportional to $a$, which has not yet been studied, couples the orientation of a rod to the relative position of another one—it favors configurations where the orientation $\mathbf{e}_\parallel[\varphi]$ is perpendicular to the axis $\Delta\mathbf{r}$ which connects the two centers of mass. We note in this context that nematic alignment of the body axis of elliptical rods does not necessarily avoid contact: two nematically aligned rods lying side-by-side may marginally overlap in contrast to the situation where one rod is behind the other one with respect to its direction of motion. Another way of looking at this new torque term is that a rod which approaches another one will turn away, thereby minimizing the potential overlap. We refer to this novel term as nematic collision avoidance. Note that distance-dependent repulsion as expected for soft spheres, though with a different symmetry, was recently considered in a Vicsek-like model with nematic alignment and velocity reversals[59], i.e., in dry active nematics, where repulsion was shown to play a central role for the emergent macroscopic patterns at high density.

**Self-propulsion breaks uniaxial nematic symmetry.** We describe the overdamped dynamics of spatially extended, active particles

via

$$\dot{\mathbf{r}}_k = v_0 \mathbf{e}_\parallel[\varphi_k] + \hat{\boldsymbol{\mu}}[\varphi_k] \cdot \sum_j \mathbf{f}_2(\mathbf{r}_k - \mathbf{r}_j, \varphi_k, \varphi_j) + \boldsymbol{\eta}_k(t), \quad (4)$$

$$\dot{\varphi}_k = \mu_\varphi \sum_j m_2(\mathbf{r}_k - \mathbf{r}_j, \varphi_k, \varphi_j) + \sqrt{2D_\varphi}\xi_k(t). \quad (5)$$

The balance of dissipative and driving force leads to stochastic motion with a mean speed $v_0$ along the long particle axis $\mathbf{e}_\parallel$ in the absence of interactions. The translational ($\hat{\boldsymbol{\mu}}$) and rotational ($\mu_\varphi$) mobilities as well as the nature of fluctuations of position and orientation, abbreviated by $\boldsymbol{\eta}_k(t)$ and $\xi_k(t)$, respectively, are determined by the anisotropy of particles and the properties of the surrounding medium. The noise terms $\boldsymbol{\eta}_k(t)$ and $\xi_k(t)$ are assumed to be Gaussian, unbiased and $\delta$-correlated in time. Fluctuations of the center of mass are generally anisotropic, even if they were of thermal origin,

$$\boldsymbol{\eta}_k(t) = \sqrt{2D_\parallel}\,\mathbf{e}_\parallel[\varphi_k]\eta_{\parallel,k}(t) + \sqrt{2D_\perp}\,\mathbf{e}_\perp[\varphi_k]\eta_{\perp,k}(t), \quad (6)$$

where the diffusion coefficients parallel and perpendicular to the rods' orientation read $D_\parallel$ and $D_\perp$.

Notably, interaction, diffusion, steric repulsion, and friction possess uniaxial (nematic) symmetry: they are invariant under the transformation $\varphi_j \to \varphi_j + \pi$ for any particle by construction. Only the self-propulsion term $v_0\mathbf{e}_\parallel[\varphi_j]$ breaks this inversion symmetry at the microscale as $\mathbf{e}_\parallel[\varphi_j] \to \mathbf{e}_\parallel[\varphi_j + \pi] = -\mathbf{e}_\parallel[\varphi_j]$. For this reason, self-propelled rods ($v_0 > 0$) are inherently different from systems without directed self-propulsion on the microscale ($v_0 = 0$), where the diffusive dynamics is invariant under inversions of the orientation vector $\mathbf{e}_\parallel$, hence distinguishing self-propelled objects from most active nematic models[68].

In the equilibrium limit, when fluctuation–dissipation relations hold and self-propulsion vanishes, Eqs. (4) and (5) describe passive rods with nematic alignment, in line with Onsager's theory[51].

**Breakdown of MIPS.** The shape of rigid self-propelled particles determines their collective behavior (Fig. 2). In the limiting case of self-propelled hard discs ($l_\parallel/l_\perp = 1$), we observe MIPS—the formation of aggregates that display hexatic order, surrounded by a disordered gas of active particles[11,12,14–17,69]. In this regime, the observed phenomenology is consistent with previously reported

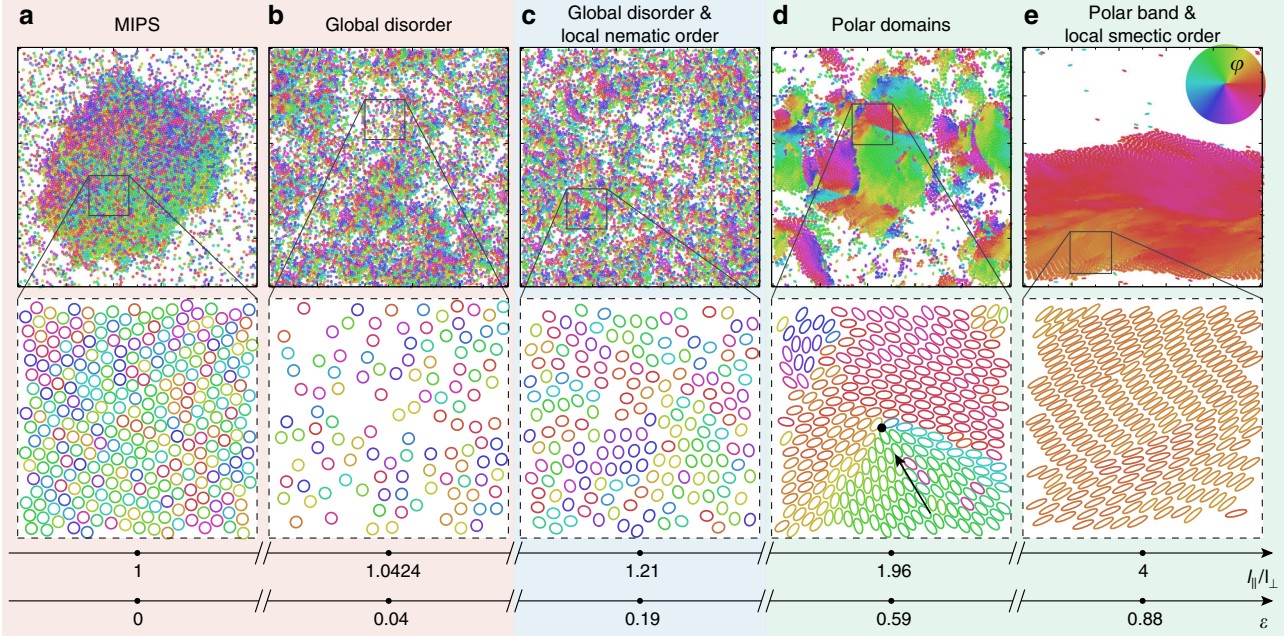

**Fig. 2 Snapshots of large-scale patterns for increasing particle anisotropy of rigid self-propelled objects.** The top row shows the entire simulation box; enlarged images (40 × 40) are plotted in the corresponding panels below. The anisotropy is varied keeping the particle size $A \propto l_{\parallel} l_{\perp}$ fixed by setting the product $l_{\parallel} l_{\perp} = 1$. From left to right: **a** $\varepsilon = 0$, aspect ratio $l_{\parallel}/l_{\perp} = 1$; **b** $\varepsilon \approx 0.04$, aspect ratio $l_{\parallel}/l_{\perp} = 1.0424$; **c** $\varepsilon \approx 0.19$, aspect ratio $l_{\parallel}/l_{\perp} = 1.21$; **d** $\varepsilon \approx 0.59$, aspect ratio $l_{\parallel}/l_{\perp} = 1.96$; **e** $\varepsilon \approx 0.88$, aspect ratio $l_{\parallel}/l_{\perp} = 4$. Color coding represents the orientation of the active force. The background shading indicates the different physical regimes. See also Supplementary Movies 1–3. Simulation parameters (cf. Methods): energy functional $\mathcal{F}[\xi] = \xi^{\gamma}$ with energy scale $\kappa = 1$ and exponent $\gamma = 3$, active force $F_a = v_0/\mu_{\parallel} = 0.01$, spatial diffusion $D_{\parallel,\perp} = 0$, rotational fluctuations $D_{\varphi} = 3 \times 10^{-5} \mu_{\varphi}$, systems size $L_{x,y} = 250$ and particle number $N = 5968$. The anisotropic mobility matrix for ellipsoids dispersed in a liquid was used[74].

findings on self-propelled discs. In Fig. 2a, the enhancement of density fluctuations in the phase-separated regime is evident. Since the orientations of discs within an aggregate is disordered, MIPS may be described by a scalar field theory for the particle density only[15].

Surprisingly, we find that MIPS aggregates melt for small anisotropies ($l_{\parallel}/l_{\perp} = 1.0424$ in Fig. 2b): in contrast to the phase-separated regime, we observe a drastic decrease of density fluctuations and a reduction of local hexatic order. Phase diagrams that quantify the behavior of the system in detail for disc-like particles as well as for slightly anisotropic rods along with measures for the level of hexatic order, both local and global, are provided in Supplementary Note 2 and Supplementary Figs. 2–4.

What is the mechanism behind the break up of aggregates? We recall that phase separation of self-driven spheres arises due to the slow down of particles as they collide[11]. Upon collision, their orientations point towards the center of clusters such that aggregates are surrounded by a polar boundary layer[70]— active pressure keeps them together (Fig. 3a). Preparing an aggregate in a slap geometry and performing a quench to slightly anisotropic rods reveals that this polar boundary layer dissolves as a deterministic torque rotates rods away from the boundary of aggregates (Fig. 3b, c).

Similar observations were reported from numerical investigations of hard rods, implemented via a Gay–Berne potential, with isotropic mobility tensor in the limit of infinite Péclet number[62]. The breakdown of MIPS aggregates was attributed in ref. [62] to a different mechanism from the one reported here, namely to a reduction of force imbalance as a consequence of local alignment. In contrast, our analysis (for theoretical considerations, see below) suggests an anisotropic coupling of density gradients to the polar order parameter field to stabilize the homogeneous state

for self-propelled rods. A decreased collision duration—implying the reduction of the hindrance that is the basis of MIPS—for rod-shaped objects due to torque had previously been described in ref. [61]. Overall, we conclude that the destabilization of MIPS is not a model-dependent phenomenon, but rather a generic feature of active systems with non-negligible torques, in line with recent experimental findings[19].

**Emergence of orientational order.** We observe the formation of states with orientational order by increasing the aspect ratio of particles beyond the breakdown of active phase separation (Fig. 2c–e). At first, the system remains globally disordered. Counterintuitively, large-scale domains with polar order are observed[32] if the anisotropy is increased further, though the interaction potential is strictly nematic. Those macroscopic patterns (Fig. 2d) are highly dynamic since polar order is inherently related to mass transport, thereby inducing clusters to form, merge and break in a nontrivial fashion[27,32].

Along with polar domains, topological defects emerge due to collisions of those structures (Fig. 2d). Examining the orientation of the rod axis only, i.e. irrespective of the orientation of the self-propulsion force with respect to the body axis, these defects have a half-integer topological charge, familiar from active nematics[68]. Defects may, however, be self-motile because of the polarity of directional energy input at the level of individual rods: in Fig. 2d, a black arrow indicates that rods push towards the center of a $+\frac{1}{2}$-defect thereby creating an active, anisotropic stress, which results in a directed displacement of the defect position. This mechanism of defect motion in ensembles of polarly driven objects is different from defect motility in active nematics, both dry[44] and wet[68]. Furthermore, defects are created and disappear in an intermittent way: due to strong density instabilities, defects

**Fig. 3 Unstable MIPS aggregate after a quench from discs to slightly anisotropic rods.** A quench is performed from circular particles (panel **a**) to ellipsoidal particles with the anisotropy $l_\parallel / l_\perp = 1.21$ (panels **b**, **c**). A cross-section of the density field and the $p_x$-component of the polar order parameter field in a slap geometry is shown together with corresponding snapshots for three different times $t_1 < t_2 < t_3$. The state shown in panel **c** is not a stationary state, which is given by a flat density profile. Color coding represents the orientation, as indicated by the color bar. Simulation parameters (cf. Methods): energy functional $\mathcal{F}[\xi] = \xi^\gamma$ with energy scale $\kappa = 1$ and exponent $\gamma = 3$, fixed particle surface area $A = \pi l_\parallel l_\perp = \pi$, active force $F_a = v_0 / \mu_\parallel = 0.01$, spatial diffusion $D_{\parallel,\perp} = 0$, rotational diffusion $D_\varphi = 7.5 \times 10^{-5}$, systems size $L_x = 300$ and $L_y = 70$, particle number $N = 1671$.

may vanish in the void or penetrate from the boundary of a dense region—the topological charge is therefore not conserved.

We observe that polar domains may become system spanning for intermediate system sizes for high anisotropy (Fig. 2e), in line with the findings reported in refs. [27,34,36]. These polarly ordered structures are furthermore comprised of smectic particle arrangements due to the nematic collision avoidance torque that favors configurations where the orientation of rods is perpendicular to the axis which connects the two centers of mass of neighboring rods [cf. the discussion of Eq. (3)]. Order parameters that quantify the local positional structure of bands are discussed in Supplementary Note 2 (Supplementary Fig. 6). Note that numerical data suggest the absence of long-range orientational order in the thermodynamic limit[34].

We stress that macroscopic order is polar, while the symmetry of the interaction potential of rods is strictly nematic (uniaxial symmetry). Accordingly, the symmetry of macroscopic order is not imposed by the symmetry of the interaction potential, but emerges spontaneously from the spatial dynamics of particles. Similar behavior was observed in studies of myxobacteria[38,71].

Our simulations reveal that the emergence of local polar order —inherently related to convective mass transport—is accompanied by strong density instabilities. However, this does not imply that MIPS is reentrant with the aspect ratio. We underline in this context the significant structural differences of the states shown in Fig. 2a, d, e. Clusters are highly dynamic in the case of self-propelled rods, and their persistence grows with cluster size. On the contrary, aggregates formed by self-propelled discs via MIPS move diffusively, with a diffusion coefficient that decreases with system size (cf. Supplementary Note 2 and Supplementary Fig. 5 for characterization of particle transport). Accordingly, the shape of clusters and their dynamics differ. Note in this context that aggregates formed via MIPS are surrounded by a polar boundary layer (Figs. 2a and 3) of particles pointing toward the center of the aggregate whereas rods are aligned parallel to the band axis (parallel to the interface) in the case of rods (Fig. 2e). In short, these patterns differ in their surface structure and dynamics (see also Supplementary Note 2 and Supplementary Fig. 6 for additional details).

**Polar vs. nematic order and their coexistence.** Generally, the emergent patterns formed by active particles differ if particles can slide past each other—implying that the self-propulsion force can overcome repulsive interactions—compared to situations where this is impossible[37]. To address this question, we now fix the aspect ratio and ask what the influence of self-propulsion for this system is by varying the speed $v_0$. In the limit of high self-propulsion, particles may slide past each other upon encounter whereas they are blocked by their interaction partners in the opposite limit. Figure 4 depicts graphically the phenomenological

transition from weakly to strongly driven rods. Large-scale, polar domains are observed for low self-propulsion. In the limit of high activity, in contrast, particles arrange themselves in nematic band-like structures as they are familiar from Vicsek-type, point-like particles with nematic alignment[24]. In a small parameter window, where the order of magnitude of the self-propulsion force is comparable to repulsive forces, a bistable coexistence region is observed: for intermediate values of the active speed, both nematic bands and polarly ordered domains are observed intermittently. The coexistence of polar and nematic order, along with the explicit speed values $v_0$ for which the respective states are found, is detailed in Fig. 4. The stochastic switching from polar to nematic states is revealed by anomalously high fluctuations of the polar order parameter (Fig. 4b). The timescales of these stochastic transitions are remarkable as they are several orders of magnitude larger than timescales of the dynamics at the particle level.

The simultaneous existence of polar and nematic states has recently been reported by Huber et al.[43] for a motility assay experiment. Those results were rationalized by simulations of self-propelled, flexible filaments which are pulled at one side and interact by a combination of polar and nematic alignment. In contrast to this motility-assay system, the interaction of the self-propelled particles considered here is strictly nematic; the global nematic symmetry is solely broken by self-propulsion. Thus, our results reveal for the first time that bistability of polar and nematic structures can nevertheless be expected for simple self-propelled rods despite purely nematic interaction symmetry, if the strength of self-propulsion and repulsion are fine-tuned or happen to coincide in a specific application. However, we do not expect the bistability of globally polar and nematic states to be retained in the thermodynamic limit as the diffusive motion of particles within the nematic band is too slow to allow for long-range nematic order[34]. We rather expect disconnected patches composed of polarly or nematically aligned particles to emerge which coarsen in a nontrivial way as they interact at the mesoscale.

**Particle-anisotropy induces nonequilibrium stresses.** Based on the Fokker–Planck equation for the one-particle density corresponding to the particle-based description (cf. Methods), we analytically address the breakdown of MIPS for small particle anisotropies (Figs. 2 and 3). As numerical simulations reveal that this instability occurs for small particle anisotropy, it is sufficient to keep leading orders in $\varepsilon$. In this limit, the force reduces to an isotropic central body force. The torque, on the other hand, possesses at leading order $\varepsilon$ a contribution that results from the novel nematic collision avoidance term

$$m_2\left(\mathbf{r}_k - \mathbf{r}_j, \varphi_k, \varphi_j\right) \propto \varepsilon \sin\left[2\left(\varphi_k - \arg\left[\mathbf{r}_k - \mathbf{r}_j\right]\right)\right]. \quad (7)$$

This interaction mechanism has not been studied analytically so far in the context of active matter to the best of our knowledge.

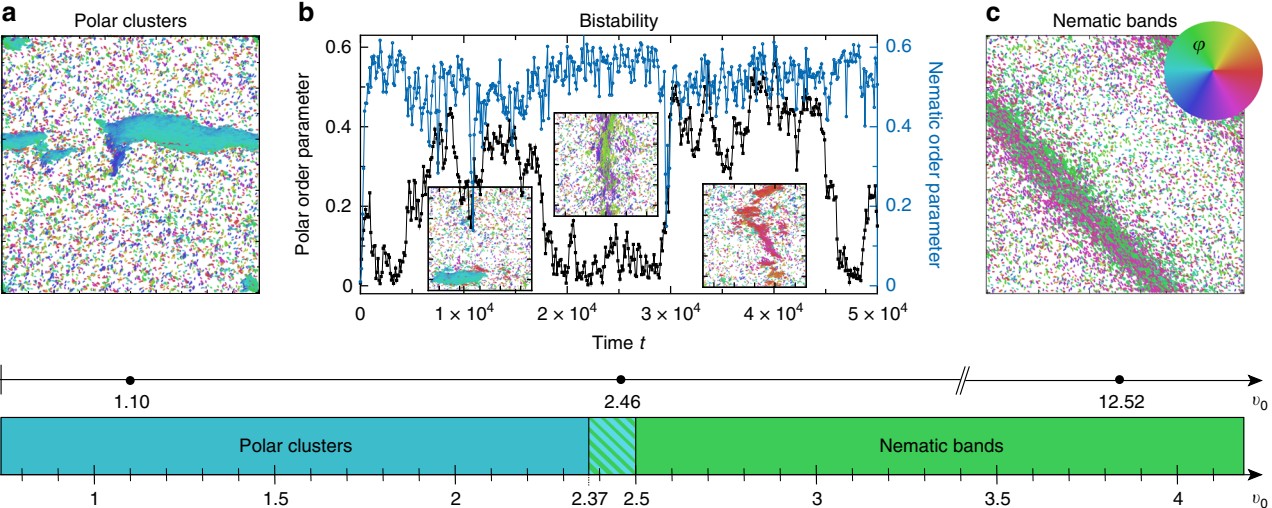

**Fig. 4 Coexistence of polar and nematic order for self-propelled rods.** The panels show snapshots of large-scale patterns observed in numerical simulations of rods as a function of their self-propulsion speed for fixed repulsion strength and fixed anisotropy. If the self-propulsion is small, such that repulsion forces cannot be overcome (panel **a**), polarly ordered domains are observed. In contrast, nematic bands—previously reported for Vicsek-type particles with nematic velocity alignment[24]—emerge as the high self-propulsion force allows particles to glide over each other (panel **c**). Surprisingly, we find a bistable coexistence of nematic bands and polar clusters for intermediate values of the self-propulsion force: the nematic order parameter $\left| \sum_{j=1}^{N} e^{2i\varphi_j} /N \right|$ fluctuates around a constant value, whereas the polar order parameter $\left| \sum_{j=1}^{N} e^{i\varphi_j} /N \right|$ switches stochastically between two values, corresponding to one state with polar order (nonzero value of polar order parameter) to a nematic state where the polar order parameter fluctuates close to zero (panel **b**); snapshots are shown as insets (see also Supplementary Movie 4). The lower panel shows the speed values $v_0$ for which the respective states are observed—bistability of polar and nematic structures, indicated by a hatched pattern, is expected for $2.37 \lesssim v_0 \lesssim 2.5$. Simulation parameters (cf. Methods): fixed particle shape $A \propto l_{\parallel} l_{\perp}$ by $l_{\parallel} l_{\perp} = 1$, anisotropy $\varepsilon \approx 0.88$, aspect ratio $l_{\parallel}/l_{\perp} = 4$, energy functional $\mathcal{F}[\xi] = \xi^{\gamma}$ with energy scale $\kappa = 1$ and $\gamma = 3$, spatial diffusion $D_{\parallel,\perp} = 0$, rotational diffusion $D_{\varphi} = 0.022$, system size $L_{x,y} = 500$ in (**a**) and (**c**), $L_{x,y} = 250$ in (**b**), particle density $\rho_0 = 0.08$.

Mobility and diffusion tensor of individual particles are simplified to be isotropic for small $\varepsilon$: $\hat{\boldsymbol{\mu}} \approx \bar{\mu} \mathbb{1}$ and $D_0 = D_{\parallel} \approx D_{\perp}$.

We begin the analysis by calculating the average force and torque felt by a particle with orientation $\varphi$ to first order in gradients

$$\mathbf{F} \simeq -\zeta_0 \kappa \mathbf{e}_{\parallel}[\varphi] \rho, \tag{8}$$

$$M \simeq -\varepsilon \zeta_1 \kappa \mathbf{e}_{\parallel}[\varphi] \wedge \nabla \rho = -\varepsilon \zeta_1 \kappa \left( \cos \varphi \, \partial_y - \sin \varphi \, \partial_x \right) \rho. \tag{9}$$

The parameters $\zeta_{0,1}$ are positive nonequilibrium transport coefficients, given by integrals over the pair correlation function, and $\rho$ is the particle density; for details of the derivation, see Methods and the Supplementary Note 3. Measurements of pair-correlations in particle-based simulations reveal that the kinetics of collisions leads to an enhancement of particle density in front with respect to the direction of motion of a focal particle (aka bulldozer effect): as particles move actively in a semi-dilute environment, they tend to collide with others, and consequently the probability to find a particle in front is higher than in the back (see Fig. 5a). This phenomenon was reported in system of self-propelled discs, where it was used to build a scalar field theory to describe MIPS[17,69].

According to Eq. (8), forces yield a speed reduction on average in high density areas as particles bump into their neighbors. The decrease of speed with particle density is the classical mechanism underlying MIPS[11,12]. In Eq. (9), we report an important novel element: a slight asymmetry of particle shape gives rise to a torque, which induces a rotation away from high density domains.

The physical effects of force and torque become evident at the level of coarse-grained order parameters: the density $\rho(\mathbf{r}, t)$ and the polar order parameter field $\mathbf{p} = \left\langle \mathbf{e}_{\parallel}[\varphi] \right\rangle$. Equations for these quantities are obtained by performing a mode expansion of the

Fokker-Planck equation for the one-particle density, which yields

$$\partial_t \rho \approx -\nabla \cdot [\nu(\rho) \mathbf{p}] + D_0 \Delta \rho, \tag{10}$$

$$\partial_t \mathbf{p} \approx -\nabla \cdot \left[ \frac{\nu(\rho)}{2} \Pi_+ \right]$$
$$-D_{\varphi} \mathbf{p} - \frac{\mu_{\varphi} \varepsilon \zeta_1 \kappa}{2} \Pi_- \cdot \nabla \rho + D_0 \Delta \mathbf{p}. \tag{11}$$

The convective term in Eq. (10) represents the density-dependent speed reduction due to collisions via $\nu(\rho) = \nu_0 - \bar{\mu} \zeta_0 \kappa \rho$. At the field level, the torque is cast as an anisotropic, nonlinear flow of the form $\dot{\mathbf{p}} \propto -\varepsilon \Pi_- \cdot \nabla \rho$ with the tensors $\Pi_{\pm} = \rho \mathbb{1} \pm \mathfrak{Q}$, where $\mathfrak{Q}(\mathbf{r}, t)$ abbreviates the nematic order parameter field. Accordingly, density gradients are counteracted by an opposing particle flow due to torque. The coupling of the polar order parameter to density gradients turns out to be a stabilizing mechanism of the homogeneous, disordered state, that is also responsible for the suppression of MIPS for self-propelled rods as argued below. That is in stark contrast to the arguments given in ref. [62], where this type of coupling is absent.

The time-independent solutions of these transport equations imply the polar order parameter $\mathbf{p}$ to be collinear to the density gradient $\nabla \rho$, as $\mathbf{p} = D_0 (\nabla \rho)/\nu(\rho)$ follows from Eq. (10), cf. the phase-separated state in Fig. 3a. The theoretical analysis for anisotropic particles reveals, however, that torques will destabilize parallel arrangements of the orientation $\mathbf{e}_{\parallel}[\varphi]$ and the density gradient $\nabla \rho$ [see Eq. (9)]. Therefore, the torque, which is proportional to the anisotropy $\varepsilon$, tends to dissolve the polar boundary layer around aggregates—it induces locally anisotropic stresses whenever density gradients and local order coexist on a coarse-grained level [Eq. (11)], as we argued before based on numerical simulations.

To substantiate these arguments, we investigated the linear stability of the spatially homogeneous, isotropic state on the basis

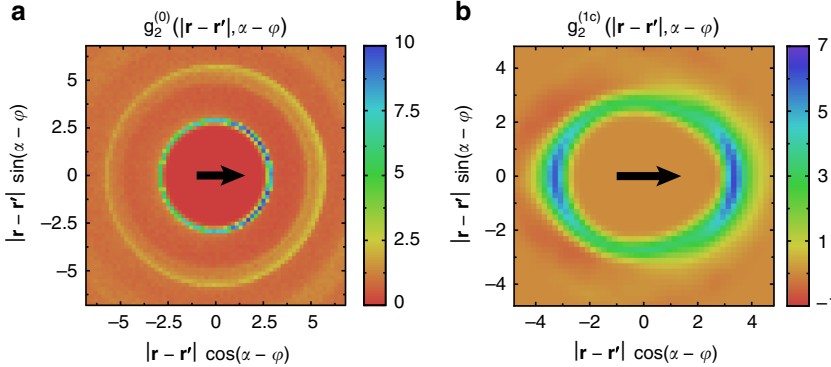

**Fig. 5 Numerical quantification of the collision kinetics in terms of the pair correlation function. a** Lowest order Fourier coefficient $g_2^{(0)}$ of the pair correlation function (cf. Methods) for spherical particles ($\varepsilon = 0$) in the disordered phase close to MIPS. The correlation function is shown in the co-moving reference frame of a particle at the origin, moving toward the right as indicated by the black arrow. If the rotational diffusion is low, the density distribution around a focal particle is asymmetric with respect to its direction of motion: the probability to find a particle in front is significantly enhanced[17]. Simulation parameters (cf. Methods): energy functional $\mathcal{F}[\xi] = \xi^\gamma$ with energy scale $\kappa = 1$ and exponent $\gamma = 3$, speed $v_0 = 0.01$, translational and rotational mobilities $\mu_\| = \mu_\perp = 1$, $\mu_\varphi = 3/4$, translational diffusion $D_{\|,\perp} = 0$, rotational diffusion $D_\varphi = 7.5 \times 10^{-4}$, systems size $L_{x,y} = 250$, particle number $N = 5968$. **b** First Fourier component $g_2^{(1c)}$ of the pair distribution function $g_2$, indicating an enhanced probability of parallel motion of close-by rods due to occasional cluster formation. The correlation function further reveals an asymmetric excluded volume in contrast to self-propelled spheres, cf. panel (**a**). Simulations were performed below the transition to local polar order ($\sigma_p < 0$). Parameters correspond to the simulation shown in Fig. 2c: $\varepsilon \approx 0.19$ and $l_\|/l_\perp = 1.21$.

of Eqs. (10) and (11), cf. Methods. Studying linear perturbations around the homogeneous state, we derive the following necessary conditions, analogous to the critical point, for the emergence of MIPS to first order in $\varepsilon$

$$v_0 > v^* = 4\sqrt{D_0 D_\varphi} + \varepsilon \mu_\varphi \zeta_1 \kappa^* \rho_0^*, \qquad (12)$$

where $\kappa^*$ and $\rho_0^*$ denote the coupling strength and density at the critical point, respectively. For $\varepsilon = 0$, this expression reduces to the well-known result for self-propelled discs[17]. Accordingly, the spinodal region within which MIPS emerges is shifted towards higher speed values. According to Eq. (12), MIPS aggregates may be restabilized by increasing the self-propulsion speed $v_0$, however, we stress that it is a necessary and not a sufficient condition for MIPS. The phase diagrams, see Supplementary Note 2 and Supplementary Fig. 4, do not show this type of restabilization, indicating that MIPS does indeed not emerge above a critical aspect ratio—there is a critical anisotropy beyond which the polar boundary layer, which would keep an aggregate together becomes unstable, in line with numerical observations (Figs. 2 and 3). These findings, particularly the dissolution of aggregates at very small anisotropies ($l_\|/l_\perp \gtrsim 1.04$), put the relevance of the classical phenomenon of MIPS for self-driven, anisotropic particles, including self-propelled rods, into question.

**Collision kinetics determines onset of orientational order.** We now examine the emergence of orientational order, as observed numerically for large anisotropies (Fig. 2). As the interaction at the particle level possesses nematic (uniaxial, front-tail) symmetry, one may naively expect the emergence of local nematic order. We observe that the break up of MIPS is indeed followed by a globally disordered phase with local nematic order. Interestingly, local order becomes, counter-intuitively, polar if the aspect ratio is increased even further. In order to identify and understand the emergence of local orientational order at the hydrodynamic level, we derived coarse-grained order parameter equations where hydrodynamic transport coefficients are expressed as integrals over the correlation functions. Here, we concentrate on central, symmetry-breaking terms for the polar and nematic order at the local level, i.e., we expand to lowest

order in spatial gradients:

$$\dot{\mathbf{p}} = \sigma_p \mathbf{p} + \mathcal{O}(\nabla), \qquad (13)$$

$$\dot{\mathfrak{Q}} = \sigma_n \mathfrak{Q} + \mathcal{O}(\nabla). \qquad (14)$$

If the transport coefficient $\sigma_p$ is positive, the local polar order parameter grows and, thus, ordered polar structures are expected at local scales. In contrast, the nematic order parameter is relevant at the local level if $\sigma_p < 0$ and $\sigma_n > 0$.

A mode expansion of the one-particle Fokker–Planck equation yields the following expressions for the relevant transport coefficients (cf. Methods)

$$\sigma_p = \frac{\rho \mu_\varphi}{2\pi} \int_0^\infty dr\, r \int_0^{2\pi} d\alpha \int_0^{2\pi} d\varphi \sin(\varphi) \tilde{m}_2(r, \alpha, \varphi) - D_\varphi, \qquad (15)$$

$$\sigma_n = \frac{\rho \mu_\varphi}{\pi} \int_0^\infty dr\, r \int_0^{2\pi} d\alpha \int_0^{2\pi} d\varphi \sin(2\varphi) \tilde{m}_2(r, \alpha, \varphi) - 4D_\varphi. \qquad (16)$$

In these equations, we introduced $\tilde{m}_2 = m_2 g_2$ which is the product of the actual torque $m_2$ between two particles and the pair distribution function $g_2$. It represents an effective mean-field model. We implicitly assumed that the pair-distribution function $g_2$ is known and absorbed it into the definition of $\tilde{m}_2$. Thereby, the transport coefficients $\sigma_{p,n}$ above still depend on the inter-particle correlations and the collision kinetics.

In mean-field approximation, where $g_2 \approx 1$, the effect of collisions and positional correlations is neglected—consequently, the effective torque $\tilde{m}_2$ is identical to the actual torque $m_2$. In this limit, the transport coefficient $\sigma_p$ for the polar order parameter is always negative; the integral in Eq. (15) vanishes for symmetry reasons: $\sigma_p = -D_\varphi$. Hence, the emergence of polar order cannot be described within mean-field theory, which can only predict the existence of an isotropic-nematic transition[60]. An in-depth comparison of Smoluchowski and Boltzmann approaches to kinetic theories for self-propelled rods also predicted the parameter $\sigma_p$ to remain negative[58] as a consequence of the mean-field approximation or the molecular chaos assumption, respectively.

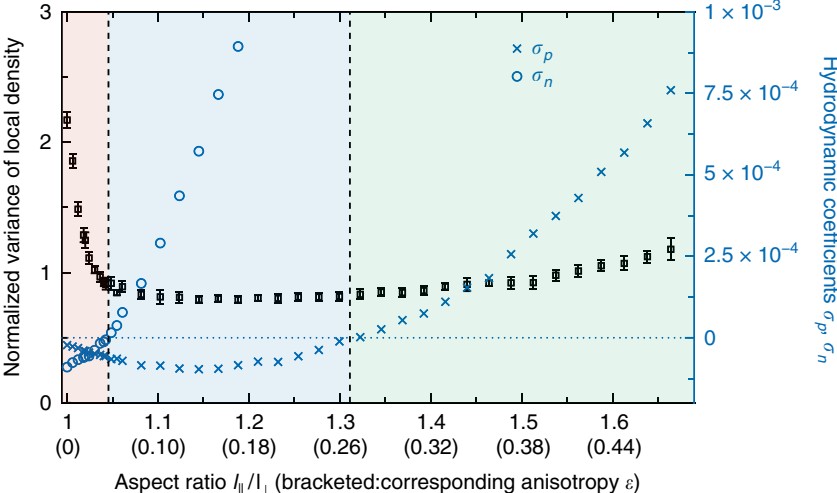

**Fig. 6 From active phase separation to collective motion by varying particle anisotropy.** At first, MIPS breaks down as density fluctuations decrease, measured in terms of the variance of the coarse-grained density field, normalized with respect to the variance which is expected in a Poissonian point pattern as a proxy of a spatially homogeneous, disordered configuration. Error bars indicate 1σ-confidence intervals. The first vertical line corresponds to the transition to local nematic order, signaled by $\sigma_n = 0$. The second vertical line indicates the onset of local polar order given by $\sigma_p = 0$ that leads to the formation of long-lived, large-scale polar clusters. The background color marks the three identified phases/regimes; for typical snapshots and parameter values, see Fig. 2.

Accordingly, the emergence of polar order for low self-propulsion is related to the collision kinetics and, in particular, to the formation of polar clusters[32] which is, in turn, reflected by correlations. In this context, we recall that only the self-propulsion force breaks the nematic symmetry of the microscopic dynamics [Eqs. (4) and (5)].

To rationalize the numerically observed emergence of polar order, we first give a heuristic argument along with the illustration of polar and anti-polar collisions in Fig. 1. We consider the limit where rods strongly repel each other such that they cannot slide past each other as active forces are too weak. Let us consider a collision under an acute angle: a rod-shaped particle colliding with a cluster aligns its direction of motion to the local mean orientation. Consequently, it will keep on moving in parallel with this cluster for a significant time. Only rotational diffusion may deflect its direction of motion away from the boundary of the cluster. Therefore, polar clusters are stable and may grow for low rotational noise. In contrast, nematic clusters cannot exist: a particle that collides in an antipolar way with a cluster will just slide off its boundary. As the distance of the particle and the cluster increases over time, their positions decorrelate for strongly repelling rods, as subsequent collisions with other particles will induce deflections of the direction of motion. Thus, we conclude that the probability to find rods moving in parallel is higher than seeing antipolar arrangements locally as a result of the collision kinetics. This can be verified quantitatively by measuring the pair correlation function numerically (Fig. 5b).

More rigorously, the effective, binary torque on the field level $\tilde{m}_2$, which enters into the relevant hydrodynamic transport coefficient $\sigma_p$, is given by the product of the model $m_2$ and the correlation function $g_2$. Hence, the effective torque $\tilde{m}_2$ depends on the kinetics of collisions. Consequently, this quantity may contain new terms with symmetries which are not present on the particle level: the torque at the particle level contains nematic alignment as $m_2 \propto \sin[2(\varphi' - \varphi)]$, and the pair distribution function contains a positive contribution $g_2 \propto \cos(\varphi' - \varphi)$ as the probability of parallel motion is enhanced. Thus, their product contains effectively positive polar alignment terms proportional to $\sin(\varphi' - \varphi)$, contributing to the first Fourier

mode $\int_0^{2\pi} d\varphi \sin(\varphi) \, \tilde{m}_2(r, \alpha, \varphi)$ in the integral in Eq. (15), though polar alignment is not explicitly present at the particle level.

In short, the presence of correlations renormalizes the interaction parameters and even introduces new interaction terms at the mesoscale. Therefore, the coefficient $\sigma_p$ can turn positive such that polar terms become relevant on the hydrodynamic level in a model with purely nematic interactions. These arguments crucially depend on the presence of self-propulsion which is the only term that breaks the global nematic symmetry. Accordingly, polar order cannot emerge in the limit $v_0 \to 0$.

We investigated numerically how the transport coefficients $\sigma_p$ and $\sigma_n$ depend on the anisotropy of particles by measuring the pair distribution function $g_2$ and evaluating the integrals in Eqs. (15) and (16). Figure 6 shows the relevant hydrodynamic coefficients together with the variance of the local density as a measure for density fluctuations. On the basis of this semi-analytical study, we distinguish the following parameter ranges, which were introduced along with Fig. 2. Negative values of $\sigma_p$ and $\sigma_n$ together with a high level of density fluctuations correspond to MIPS as observed for isotropic particles. Increasing the anisotropy, density fluctuations decrease rapidly as MIPS aggregates break up, while $\sigma_p$ and $\sigma_n$ are negative (local disorder). Subsequently, $\sigma_n$ turns positive signaling local nematic order, followed by the emergence of local polar order when $\sigma_p$ becomes positive.

**Symmetries of ordered states are emergent properties.** The symmetry of emergent patterns is essentially determined by the spatial dynamics, namely whether rods can possibly slip past each other or not[37], cf. Fig. 4. If self-propulsion forces can overcome repulsion, one can simplify Eq. (4) to $\dot{\mathbf{r}}_k \simeq v_0 \mathbf{e}[\varphi_k]$ such that the original rod model reduces to a Vicsek-type model with nematic alignment[24,37,66]—the fact that particles push each other is of minor importance in this parameter regime. Hence, the following phenomenology is expected at low density: for low rotational diffusion, a spatially homogeneous, nematic phase emerges at the mesoscale (finite system size); by increasing the noise, the level of nematic order decreases; close to the order–disorder transition, the system demixes into a high density region which is nematically ordered and a low-noise area where particles move in a disordered

fashion. This reasoning explains the type of patterns observed in numerical simulations as shown on the right of Fig. 4. We thus conclude that the nematic alignment term in the torque [Eq. (3)] dominates the large-scale dynamics in the limit of high activity. Accordingly, corresponding mean-field theories for Vicsek-type self-propelled rods[57,67] account for the observed pattern formation phenomena, such as band formation and nematic ordering. This implies that positional correlations are less relevant if particles move fast as the system becomes well-mixed, i.e., particle positions decorrelate quickly when rods can slip over each other.

By decreasing the self-propulsion force or, equivalently, increasing the repulsion strength, particles would, however, get blocked upon encounter, positional correlations build up and the pair-correlation function becomes increasingly relevant such that mean-field arguments are not applicable. In this regime, the symmetry of emerging patterns may differ from the symmetries of the microscopic interaction. It is an open challenge for future work beyond the present study to derive the pair correlation function of anisotropic, self-propelled objects from first principles—analogues to corresponding theories for self-propelled discs[69] —including nonlinear cross-coupling terms in the evolution of nematic and polar order parameters[57]. The full account of the emergent bistability of polar and nematic order would require to show how it is possible to observe nematic order, polar order or their coexistence, as all of these situations are possible for the studied system of anisotropic, self-propelled particles. The magnitudes of the nonlinear transport coefficients control whether nematic or polar order prevails, or both coexist. This limit is most difficult to assess analytically as standard series expansions fail and, moreover, density instabilities and orientational order are intrinsically linked such that a theoretical description in terms of scalar quantities only, such as the particle density in the case of MIPS of self-propelled discs, is not applicable for self-propelled rods once vectorial or tensorial order parameters grow at the local level.

## Discussion

In this novel modeling approach to active matter, individual particles are represented by smooth fields, and their interactions are derived from the minimization of energy that is a function of the overlap between particles. Force and torque are analytically obtained in contrast to rule-based algorithms, thereby enabling both analytical investigations and a convenient numerical implementation. Importantly, this modeling technique enables studying the transition from self-propelled discs[12], whose behavior is reproduced consistently, to self-propelled rods[27] by performing continuous deformations of the shape. Here, we show numerically and analytically how aggregates of circular particles formed via MIPS become unstable for weakly anisotropic, self-propelled objects. Specifically, the combined action of anisotropic repulsion and self-propulsion leads to the emergence of an effective torque, which—above a critical aspect ratio—dissolves the polar boundary layer required to maintain motility-induced aggregates. These findings provide an understanding of the role played by particle anisotropy regarding the robustness of active phase separation described in terms of scalar field theories for the particle density. Our smooth-particle approach underpins the restabilization of the disordered, homogeneous phase—respectively, the breakdown of MIPS—to specific microscopic interaction mechanisms, which, as we argue, are also involved in the emergence of order, moving beyond previous studies that reported the destabilization of MIPS based on numerical measurements of effective transport coefficients[61] or identified a different destabilization mechanism[62].

Furthermore, we show within the same theoretical framework, that both, aspect ratio as well as the ratio of rigidity and self-propulsion, control the symmetry of the pair correlation function, in turn determining the onset of orientational order. Importantly, the emerging order can be either of nematic or polar nature and is therefore not dictated by the symmetry of the interaction potential only. It depends on the emergent properties of the pair correlation function. That is why polar and nematic structures can simultaneously coexist in a system of identical particles with purely nematic interactions. Our analysis reveals that the symmetry of macroscopic order is an emergent and dynamic property of active systems, similar to recent findings from the analysis of a motility assay experiment[43]. Thus, both polar and nematic order parameters shall be taken into account on the hydrodynamic level.

In summary, the developed framework enables studying MIPS for isotropic, self-propelled particles, its breakdown with particle anisotropy, as well as the emergence of both polar and nematic order, and their coexistence, for the same type of active particles. Thus, this framework provides a comprehensive picture of most relevant phenomena reported for active systems and, thereby, contributes to linking scalar to vectorial active matter[37,61,62]. Therefore, we expect our framework to shed light on a large number of applications, including the growth of bacterial colonies or self-organized patterns in systems of active filaments. Furthermore, it could be utilized to assess the role of the particle density in active matter, particularly for bulk phases close to the percolation threshold. The simplicity of the proposed model may also help to address the highly nontrivial effect of hydrodynamic interactions on the collective dynamics of active systems. In particular, it may provide insight into the question of different routes to pattern formation via a blocking effect, comparable to self-propelled discs[12], or alignment-induced clustering of rod-shaped particles[27]. Pioneering works in this direction indicate how the alignment of self-propelled rods changes due to hydrodynamic flows that may act synergistically or antagonistically, depending on the type of swimmer (pusher vs. puller)[21,72]. Furthermore, natural extensions of the developed approach range from the addition of spatially disordered environments to the study of polydisperse systems or semi-flexible, filamentous particles[73], among many others.

## Methods

**Interaction energy, force, and torque.** Each particle is represented by an anisotropic Gaussian field

$$\psi_k(\mathbf{r}) = e^{-\frac{\left\{(\mathbf{r}-\mathbf{r}_k)\cdot\mathbf{e}_\parallel[\varphi_k]\right\}^2}{2l_\parallel^2} - \frac{\left\{(\mathbf{r}-\mathbf{r}_k)\cdot\mathbf{e}_\perp[\varphi_k]\right\}^2}{2l_\perp^2}}. \tag{17}$$

The overlap $\mathcal{I}_{kj} = \int d^2 r\, \psi_k(\mathbf{r})\psi_j(\mathbf{r})$ of two particles can be calculated analytically:

$$\mathcal{I}_{kj} = \mathcal{I}_0(\varphi_k, \varphi_j)\, e^{-\frac{(\mathbf{r}_k - \mathbf{r}_j)\cdot\left[1 - \frac{\varepsilon}{2}\left(\mathcal{Q}[\varphi_k] + \mathcal{Q}[\varphi_j]\right)\right]\cdot(\mathbf{r}_k - \mathbf{r}_j)}{2\left[1 - \varepsilon^2\cos^2\left(\varphi_k - \varphi_j\right)\right]\left(l_\parallel^2 + l_\perp^2\right)}}, \tag{18}$$

where $\mathcal{Q} = \mathbf{e}_\parallel \otimes \mathbf{e}_\parallel - \mathbf{e}_\perp \otimes \mathbf{e}_\perp$ and

$$\mathcal{I}_0 = \mathcal{I}_0\left(\varphi_k, \varphi_j\right) = \pi l_\parallel l_\perp \sqrt{\frac{1 - \varepsilon^2}{1 - \varepsilon^2\cos^2\left(\varphi_k - \varphi_j\right)}}. \tag{19}$$

The fields $\psi_k \in [0, 1]$ are not probability distribution functions, but shall rather indicate where a rod is located in space, similar to a phase-field. Therefore, Eq. (17) is not normalized like a Gaussian probability density. The normalization of $\psi_k$ is rather chosen such that the surface overlap of two particles with identical orientations, $\varphi_j = \varphi_k$ and identical centers of mass $\mathbf{r}_j = \mathbf{r}_k$ is equal to the area of an ellipse in two dimensions: $\mathcal{I}_{kl} = \pi l_\parallel l_\perp$.

The interaction energy $\mathcal{U}$ is defined as the sum of binary contributions: $\mathcal{U} = \frac{1}{2}\sum_{k,j}^{N} u_2\left(\mathbf{r}_k - \mathbf{r}_j; \varphi_k, \varphi_j\right)$. For passive systems, where strong overlapping rarely occurs, the binary interaction energy $u_2$ can directly be defined as an increasing function of the overlap $\mathcal{I}_{kj}$[65]. For active systems, however, or in contexts where strong particle overlapping cannot be ignored, the dependency on $\mathcal{I}_0$ has to be discarded to ensure that the same interaction symmetry is maintained at all densities (see Supplementary Note 1 and Supplementary Fig. 1 for a detailed

discussion). Thus, we define $u_2$ as

$$u_2(\Delta\mathbf{r};\varphi,\varphi') = \kappa\mathcal{F}\left[ e^{-\frac{\Delta\mathbf{r}\cdot[1-\frac{\varepsilon}{2}(\mathcal{Q}[\varphi]+\mathcal{Q}[\varphi'])]\cdot\Delta\mathbf{r}}{2[1-\varepsilon^2\cos^2(\varphi-\varphi')]\left(l_\parallel^2+l_\perp^2\right)}} \right], \tag{20}$$

where $\Delta\mathbf{r} = \mathbf{r}' - \mathbf{r}$ is the relative position, $\kappa$ is the interaction strength measured in units of energy and $\mathcal{F}[\xi]$ is a monotonically increasing function of the overlap. In this way, the energy increases as particles approach each other, hence inducing a repulsive force. In particular, soft and hard objects can be described: if the energy is finite for $\Delta\mathbf{r} \to 0$, particles are soft whereas these objects can be considered hard if the energy diverges in this limit.

The binary force $\mathbf{f}_2(\Delta\mathbf{r},\varphi,\varphi') = -\nabla u_2(\Delta\mathbf{r},\varphi,\varphi')$ and torque $m_2(\Delta\mathbf{r},\varphi,\varphi') = -\partial_\varphi u_2(\Delta\mathbf{r},\varphi,\varphi')$ exerted on a particle located at $\mathbf{r}$ with orientation $\varphi$ by another one at $\mathbf{r}'$ with orientation $\varphi'$ are deduced from the potential energy by differentiation with respect to its position and orientation, respectively. Mathematical details of this derivation are provided in Supplementary Note 1.

**Numerical Langevin simulations.** The numerical integration of the Langevin dynamics was performed via a stochastic Euler scheme. The interaction was simplified for numerical purposes: the force and torque decay exponentially and are thus practically zero beyond interparticle distances that are much larger than the typical decay length. Therefore, we neglect interactions of particles which are separated by more than five standard deviations, in terms of the characteristic Gaussian decay of the interaction energy with particle separation $\Delta\mathbf{r}$.

The number of model parameters can be reduced by identifying the intrinsic scales of the system. The mass scale $\mathfrak{M}$ is determined by the mass of individual particles which is of minor relevance in the overdamped limit. As an intrinsic length scale $\mathfrak{L}$ we choose the geometric mean of $l_\parallel$ and $l_\perp$: $\mathfrak{L} = \sqrt{l_\parallel l_\perp}$. A third independent parameter is the energy scale $\mathfrak{E} = \kappa$ of the binary interaction [Eq. (20)]. Thus, the intrinsic timescale is determined by $\mathfrak{T} = \mathfrak{L}\sqrt{\mathfrak{M}/\mathfrak{E}}$ and velocities are measured in multiples of the intrinsic value $\mathfrak{B} = \sqrt{\mathfrak{E}/\mathfrak{M}}$; scales for noise amplitudes and mobilities follow accordingly. Throughout, we use dimensionless quantities by rescaling time, length, and mass such that $\kappa = 1$ and the area $A = \pi$ of individual rods are fixed.

**Kinetic theory.** The main observable linking particle-based descriptions such as Eqs. (4) and (5) and a field theoretical treatment is the one-particle density distribution

$$P(\mathbf{r},\varphi,t) = \left\langle \sum_{j=1}^{N} \delta\big(\mathbf{r} - \mathbf{r}_j(t)\big)\delta\big(\varphi - \varphi_j(t)\big)\right\rangle, \tag{21}$$

which determines the density of particles at a particular reference point in phase space $\{\mathbf{r},\varphi\}$. Its dynamics

$$\partial_t P = -\nabla\cdot[(v_0\mathbf{e}[\varphi] + \hat{\boldsymbol{\mu}}[\varphi]\cdot\mathbf{F})P] + \nabla\cdot[\mathcal{D}[\varphi]\cdot\nabla P] \\ - \partial_\varphi[\mu_\varphi MP] + D_\varphi\partial_\varphi^2 P, \tag{22}$$

is systematically derived from the $N$-particle Fokker-Planck equation corresponding to the particle-based dynamics. In general, however, the equation of the one-particle density is not closed but it depends on the pair correlation function $g_2$ via the force and torque functionals:

$$\mathbf{F} = \int d^2r' d\varphi' \mathbf{f}_2(\mathbf{r}-\mathbf{r}',\varphi,\varphi')P(\mathbf{r}',\varphi',t)g_2(\mathbf{r},\mathbf{r}';\varphi,\varphi',t), \tag{23}$$

$$M = \int d^2r' d\varphi' m_2(\mathbf{r}-\mathbf{r}',\varphi,\varphi')P(\mathbf{r}',\varphi',t)g_2(\mathbf{r},\mathbf{r}';\varphi,\varphi',t). \tag{24}$$

All theoretical considerations in this work are based on this nonlinear Fokker-Planck equation for the one-particle density $P(\mathbf{r},\varphi,t)$. Note that the binary interaction $\mathbf{f}_2$ and $m_2$, i.e., the force and torque, which one particle exerts on another interaction partner [cf. Eq. (3)], are effectively renormalized by the emergent correlations quantified by $g_2$.

**Coarse-grained order parameters.** Both, in the context of the breakdown of MIPS and for the emergence of orientational order, we considered a moment expansion of the Fokker–Planck equation (22). This is generally done by temporal differentiation of the respective order parameter

$$\partial_t \rho(\mathbf{r},t) = \int_{-\pi}^{\pi} d\varphi\, \partial_t P(\mathbf{r},\varphi,t) \tag{25}$$

$$\partial_t \mathbf{p}(\mathbf{r},t) = \int_{-\pi}^{\pi} d\varphi\, \mathbf{e}_\parallel[\varphi]\partial_t P(\mathbf{r},\varphi,t), \tag{26}$$

$$\partial_t \mathfrak{Q}(\mathbf{r},t) = \int_{-\pi}^{\pi} d\varphi\, \mathcal{Q}[\varphi]\partial_t P(\mathbf{r},\varphi,t), \tag{27}$$

and insertion of the Fokker–Planck equation (22) on the right hand side. These order parameters are directly related to the Fourier modes of $P(\mathbf{r},\varphi,t)$ with respect to the angular variable $\varphi$, thereby enabling a more direct calculation of the relevant transport coefficients using Fourier transform. Technical details of the derivation of Eqs. (13)–(16) are summarized in Supplementary Note 3.

**Necessary condition for MIPS.** For self-propelled spheres, the emergence of MIPS is signaled by a long-wavelength instability of the density field[17]. Here, we examine the stability of the isotropic state with respect to long-wavelength perturbations for anisotropic particles. For this purpose, the dynamics of the polar order parameter field is linearized first by inserting $\rho = \rho_0 + \delta\rho$ and $\mathbf{p} = \delta\mathbf{p}$. As we are interested in the onset of a long-wavelength instability, the linearized field $\delta\mathbf{p}$ can further be adiabatically eliminated yielding

$$\delta\mathbf{p} \simeq -\frac{(v_0 - 2\bar{\mu}\zeta_0\kappa\rho_0) + \varepsilon\mu_\varphi\zeta_1\kappa\rho_0}{2D_\varphi} \nabla\delta\rho. \tag{28}$$

To leading order, one thus obtains an effective diffusion equation $\partial_t\delta\rho \simeq \Gamma\Delta\delta\rho$ for the fluctuations of the density around the spatially homogeneous state by inserting this expression into Eq. (10), where the transport coefficient $\Gamma$ reads

$$\Gamma = D_0 + \frac{(v_0 - \bar{\mu}\zeta_0\kappa\rho_0)\left[(v_0 - 2\bar{\mu}\zeta_0\kappa\rho_0) + \varepsilon\mu_\varphi\zeta_1\kappa\rho_0\right]}{2D_\varphi}. \tag{29}$$

A long-wavelength instability of the homogeneous state towards a phase-separated regime occurs for $\Gamma < 0$. Following the procedure presented in ref. [17] for self-propelled discs, we reformulate the instability condition $\Gamma < 0$ into the form of a quadratic equation

$$(2\bar{\mu}\zeta_0\kappa\rho_0)^2 - \mathfrak{p}(2\bar{\mu}\zeta_0\kappa\rho_0) + \mathfrak{q} < 0, \tag{30}$$

where $\mathfrak{p} = 3v_0 + \varepsilon\mu_\varphi\zeta_1\kappa\rho_0$ and $\mathfrak{q} = 2(v_0^2 + 2D_0 D_\varphi + \varepsilon\mu_\varphi\zeta_1\kappa v_0\rho_0)$. The instability region does only exist for $\mathfrak{p}^2 - 4\mathfrak{q} > 0$; otherwise there are no physical parameters satisfying Eq. (30). This inequality yields the condition discussed in the main text [Eq. (12)].

Beyond the identification of the transition line towards MIPS, we note that Eq. (29) could, in principle, be used to numerically quantify the transport properties of the system, as it represents the collective diffusion coefficient of the density field[61]. However, the validity of its derivation requires density fluctuations to be small (expansion to linear order in $\delta\rho$) and the dynamics of local order parameters to be fast and enslaved to the density (adiabatic elimination of $\delta\mathbf{p}$). Therefore, it can be used for homogeneous states with diffusive transport in the absence of local order and, thus, its applicability is limited in the context of self-propelled rods. That is why we show the mean-squared displacement of particles in Supplementary Note 2, a measure that quantifies the transport properties, beyond the disordered states with diffusive transport, applicable to other collective states emerging in self-propelled rods with local polar ordering.

**Pair distribution functions.** Our theoretical arguments are based on the enhancement of the probability to find interaction partners in front with respect to the direction of self-propulsion. This is reflected by the pair distribution function $g_2$, specifically by the Fourier component

$$g_2^{(0)}(|\mathbf{r}-\mathbf{r}'|,\arg(\mathbf{r}'-\mathbf{r})) = \frac{1}{2\pi}\int_{-\pi}^{\pi} d\varphi\, g_2(|\mathbf{r}-\mathbf{r}'|,\arg(\mathbf{r}'-\mathbf{r}),\varphi) \tag{31}$$

as shown in Fig. 5a for spherical particles, cf.[17,69].

We argued further that the probability of parallel motion is enhanced on average as a consequence of the anisotropic body shape[32], see also the schematic collisions in Fig. 1. This is reflected by the pair correlation function. In particular, a positive contribution to the Fourier component

$$g_2^{(1c)}(|\mathbf{r}-\mathbf{r}'|,\arg(\mathbf{r}'-\mathbf{r})) = \frac{1}{\pi}\int_{-\pi}^{\pi} d\varphi\, \cos(\varphi)\, g_2(|\mathbf{r}-\mathbf{r}'|,\arg(\mathbf{r}'-\mathbf{r}),\varphi) \tag{32}$$

reflects that the probability of moving together in groups ($\varphi \approx \varphi'$) is larger than moving in an anti-parallel fashion ($\varphi \approx \varphi' + \pi$). This argument has been verified by numerical measurements of the respective part of the pair distribution function (Fig. 5b).

## Data availability
The data that support the findings of this study are available from the corresponding author upon reasonable request.

## Code availability
The computer codes used for simulations and numerical calculations are available from the corresponding author upon reasonable request.

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

## Acknowledgements

R.G. and F.P. acknowledge support from the Agence Nationale de la Recherche via Grant no. ANR-15-CE30-0002-01. R.G. acknowledges funding from the People Programme (Marie Curie Actions) of the European Union's Seventh Framework Programme (FP7/2007-2013) under REA grant agreement n. PCOFUND-GA-2013-609102, through the PRESTIGE programme coordinated by Campus France. ISA was supported by the NSF PHY-1707900.

## Author contributions

F.P. conceived the project. R.G. derived the modeling framework, carried out numerical simulations, and performed the calculations. F.P., R.G., and I.S.A. discussed and interpreted the results. R.G., F.P., I.S.A. wrote the paper and commented on the paper.

## Competing interests

The authors declare no competing interests.
