## [Peer Review File · Nature Communications]

REVIEWER COMMENTS

Reviewer #1 (Remarks to the Author):

This paper proposes a theoretical framework to understand the collective behavior of two-dimensional self-propelled ellipsoidal particles with tunable aspect ratio. By considering individual active particles as spatially extended objects represented by continuum fields, the authors propose an interaction energy based on particle overlap, which allows them to obtain analytical expressions for the interaction forces and torques between particles. The authors then perform simulations and obtain analytical predictions for the phase behavior of the system taking both positional and orientational interactions into account. Thereby, this study provides a connection between the phenomenology of non-aligning and aligning active matter. The authors reach several major conclusions:

- 1 - Very small particle-shape anisotropy is enough to prevent motility-induced phase separation (MIPS), a landmark active-matter phenomenon in systems of self-propelled disks. The breakdown of MIPS is due to interparticle torques and associated non-equilibrium fluxes.
- 2 - Emergence of polar order (polar domains and bands) even when interparticle interactions are nematic, implying that the symmetry of ordered domains is an emergent property of the dynamics.
- 3 - Self-propulsion speed and the interaction energy scale control whether the system forms polar clusters, nematic bands, or a coexistence of both types of local order, which might help explain recent experimental observations.

Importantly, the authors provide theoretical insights into their observations. The theoretical calculations show how new interparticle torques destabilize MIPS, and how correlations that result from particle collisions determine the onset of local orientational order in a way that requires going beyond mean-field descriptions.

Even though bits and pieces of the reported findings were already known, a systematic theoretical framework that can explain several phenomena at once was not available. Moreover, the results could help understand phenomena observed in several experimental systems. Therefore, this manuscript represents a very valuable contribution and an important step forward in the field of active matter. The explanations are clear and the paper is beautifully written. However, as I detail below, I missed discussions of several points that would help to compare the findings of the present manuscript to previous results. These comparisons are important to assess the degree of novelty of the reported findings, and to identify the key physical ingredients that distinguish the new results from previous ones, which would strengthen the paper. Overall, I'm inclined to recommend publication if the authors successfully address the following comments and questions.

Major comments and questions:

Nematic collision avoidance

- 1 - Even though I understand how the new collision avoidance torque is derived from the interaction energy, I don't understand its physical mechanism. Can the authors explain it? The readers would benefit from a more mechanistic explanation of this torque.
- 2 - As a follow-up question, I wonder if and how the new collision avoidance torque accounts for the following situation: Imagine that particle 2 collides with particle 1 ahead of particle 1's center of mass. In this case, I understand that particle 1 should rotate away from the relative position of particle 2. However, imagine that particle 2 collides behind particle 1's center of mass. In this case, I would expect particle 1 to rotate toward the relative position of particle 2. Can the new collision avoidance torque account for this effect? If not, shouldn't this effect be accounted for?
- 3 - Given that it seems to be a generic feature of interacting rods, why wasn't the nematic collision avoidance torque accounted for in previous studies?

Breakdown of MIPS

4 - On page 4, the authors explain that, even for small particle anisotropies, MIPS aggregates melt, and that hexatic order is reduced. However, besides the simulation snapshots in Fig. 2, that authors should show data supporting these claims, for example providing some measure of the presence/absence of MIPS.

5 - How can the state of the system be described at small particle anisotropies, once MIPS clusters break down? Is there any type of order parameter, correlation, etc., that can help characterize the collective behavior of the system (besides its general disorder)?

6 - The authors explain the breakdown of MIPS for slightly anisotropic particles as a result of interparticle torques. However, as they also explain, Ref. 69 recently proposed a different mechanism for very similar observations. How are these two explanations reconciled?

7 - Similarly, van Damme et al. J. Chem. Phys. 150, 164501 (2019) reports very similar conclusions about the suppression of MIPS by interaction torques between anisotropic particles. This paper seems to be closely related to the present manuscript. The authors should therefore also discuss the similarities and differences between their model and results and those of van Damme et al.

8 - In general, what are the main novelties of the present manuscript with respect to Ref. 69 and van Damme et al.?

Emergence of orientational orders

9 - Later on, the authors explain that local nematic order appears at small particle anisotropies for states with global disorder, and that polar order appears at higher aspect ratios, as eventually explained in Fig. 6. Again, even though the snapshots in Fig. 2 are very indicative, providing quantifications would be very useful, especially when comparing to other works.

10 - Can the smectic order in Fig. 2e be quantified? Can we understand its emergence within the proposed theoretical framework?

11 - Several papers, including Refs. 33 and 36 report regimes of phase separation for self-propelled rods. Is that consistent with the findings of the present manuscript, which suggest that very small shape anisotropies readily destroy MIPS? If so, how?

12 - Along the same line, are the states in Fig. 2d-e phase-separated? Should this kind of phase separation be understood differently than MIPS? Or is MIPS reentrant with aspect ratio?

13 - To address the two points above, it might be very insightful to show the collective diffusion coefficient Γ as a function of the aspect ratio, so that instabilities of the density field can be directly compared to instabilities of the orientational fields.

Minor comments and questions:

14 - The text around Eq. 1 suggests that the particle anisotropy ϵ and the aspect ratio are independent parameters. Aren't they directly related?

15 - Should one worry about the normalization of the Gaussian field that describes the particles? If Eq. S1 were normalized, the normalization factor would depend on the aspect ratio, and this dependence would carry over to the overlap, Eq. S3. So, which one is the physically correct overlap, the one with normalized ψ or the one without the normalization factor?

16 - The Methods section is referenced right before Eq. 11, suggesting that there should be a derivation of Eq. 11 in the Methods. However, I could not find the derivation in the Methods. Is it missing?

17 - It would be helpful to provide details on the non-dimensionalization of the equations and the corresponding simulation units.

Reviewer #3 (Remarks to the Author):

This manuscript reports results of a detailed numerical and analytical study of anisotropic self-propelled particles with soft repulsion. The main focus of the manuscript is the investigation of the effect of elongation on the type of collective behavior -- from motility induced phase separation for spheres to polar motile clusters for ellipsoids.

A theoretical framework for representing active particles by continuum fields is derived.

The mechanism of destabilization of motility-induced phase separation (MIPS) and the facilitation of orientational ordering, by varying particle shape from circular to ellipsoidal, due to nonequilibrium stresses acting among self-propelled rods is elucidated. Thus, the theory bridges the gap between scalar and vectorial active matter. Depending on the parameters, like velocity and elongation, polar and nematic order may emerge and even coexist. Accordingly, the symmetry of ordered states is found to be a dynamical property of active matter.

The different collective behaviors of spherical and elongated self-propelled particles have been puzzling for quite a long time. The current study now clarifies how these differences arise, and quantifies the crossover from one to the other. Also very interesting is the coexistence of nematic and polar under certain conditions, which underlines similar observations of Ref. [42]. Therefore, I think that the manuscript is very interesting, and advances the field significantly.

The authors should consider the following comments and questions:

(1) A new model with anisotropic soft Gaussian interactions between the particles is introduced.

The collective behavior of this model is then studied by particle-based simulations, in particular for small anisotropy. As far as I can see, the results are consistent with the results of previous simulations of hard discs and rods. I think this should be stated explicitly.

(2) Fig. 4 shows the coexistence of polar and nematic order for self-propelled rods. In which region of the parameter space can such a behavior be expected and observed?

(3) I find Eq. (9) very interesting, as it shows that MIPS is possible also for elongated particles if the propulsion velocity is high enough. Is this consistent with simulations?

(4) The authors argue that the pair correlation function favors parallel (in contrast to antiparallel) alignment for large anisotropy, see Fig. 6. How can this picture be made compatible with the bistability in Fig. 4, i.e. how can nematic bands be stabilized.

It seems to me that particle density must be an essential additional variable. A few remarks should be helpful.

(5) The importance of hydrodynamic flows and interactions is mentioned both in the introduction and the discussion. In fact, there is already a simulation study of the interplay of particle elongation and hydrodynamic interactions, see

M. Theers et al., *Soft matter* 14, 8590 (2018).

The relation of this study to the current work should be briefly discussed.

Response to the reviewer 1

[Comments by the reviewer in black, our reply in blue]

This paper proposes a theoretical framework to understand the collective behavior of two-dimensional self-propelled ellipsoidal particles with tunable aspect ratio. By considering individual active particles as spatially extended objects represented by continuum fields, the authors propose an interaction energy based on particle overlap, which allows them to obtain analytical expressions for the interaction forces and torques between particles. The authors then perform simulations and obtain analytical predictions for the phase behavior of the system taking both positional and orientational interactions into account. Thereby, this study provides a connection between the phenomenology of non-aligning and aligning active matter. The authors reach several major conclusions:

1. Very small particle-shape anisotropy is enough to prevent motility-induced phase separation (MIPS), a landmark active-matter phenomenon in systems of self-propelled disks. The breakdown of MIPS is due to interparticle torques and associated non-equilibrium fluxes.
2. Emergence of polar order (polar domains and bands) even when interparticle interactions are nematic, implying that the symmetry of ordered domains is an emergent property of the dynamics.
3. Self-propulsion speed and the interaction energy scale control whether the system forms polar clusters, nematic bands, or a coexistence of both types of local order, which might help explain recent experimental observations.

Importantly, the authors provide theoretical insights into their observations. The theoretical calculations show how new interparticle torques destabilize MIPS, and how correlations that result from particle collisions determine the onset of local orientational order in a way that requires going beyond mean-field descriptions.

Even though bits and pieces of the reported findings were already known, a systematic theoretical framework that can explain several phenomena at once was not available. Moreover, the results could help understand phenomena observed in several experimental systems. Therefore, this manuscript represents a very valuable contribution and an important step forward in the field of active matter. The explanations are clear and the paper is beautifully written. However, as I detail below, I missed discussions of several points that would help to compare the findings of the present manuscript to previous results. These comparisons are important to assess the degree of novelty of the reported findings, and to identify the key physical ingredients that distinguish the new results from previous ones, which would strengthen the paper. Overall, I'm inclined to recommend publication if the authors successfully address the following comments and questions.

We thank Reviewer 1 for his/her interest in our work and the very positive feedback. These suggestions helped us to strengthen the manuscript which was thoroughly revised accordingly. Below are point-to-point answers to the Reviewer's comments.

Major comments and questions

Nematic collision avoidance

1. Even though I understand how the new collision avoidance torque is derived from the interaction energy, I don't understand its physical mechanism. Can the authors explain it? The readers would benefit from a more mechanistic explanation of this torque.

First, we point out that the novel torque contribution is a direct consequence of the overlap-minimization of two particles. We highlight in this regard that the nematic alignment of the body axis by itself does not necessarily ensure the minimization of the overlap of two rod-shaped objects. To ensure the minimization of the overlap, the vector associated to the relative position of the two rods has to be taken into consideration. We illustrate this point in the following. Let us consider two extreme cases as an illustration: two rods at a fixed distance d , which are nematically aligned as shown in the Figure below. Though the distance between the centers of mass is identical in (A) and (B) and the body axes are nematically aligned in both cases, there exists an overlap in configuration (B), highlighted in color, which is not present in situation (A). Hence, for two rods whose centers of mass are placed along the x-axis, it is more beneficial to align the body axis parallel to the y-axis (A) in contrast to the configuration (B). This argument reveals that there must exist a torque that couples the relative orientation and relative position of two rods. This very effect is what the new collision avoidance torque is reflecting.

Figure 1 The overlap of two nematically aligned rods vs. their relative positions.

To substantiate the above argument, we consider the fixed points of Eq. (3), neglecting the anisotropy of the coefficients a and b for simplicity. The second term, proportional to b , represents the nematic alignment that favors uniaxial orientation of the body axis of a pair of rods ($\varphi' = \varphi + n\pi$ with $n \in \mathbb{Z}$). The new torque contribution favors

configurations where the orientation of the body axis of a rod aligns perpendicular to the vector which connects the two centers of mass, a situation that corresponds to sketch (A) above ($\varphi = \alpha + \pi/2 + n\pi$, with $\alpha = \arg(\Delta\mathbf{r})$ and $\Delta\mathbf{r}$ the relative position vector). Thus, the configuration (A) is a stable fixed point with respect to novel *nematic collision avoidance* term whereas (B) is an unstable fixed point. Note that the *classical nematic alignment term* does not distinguish these two situations.

Furthermore, the novel contribution to the torque is a rather general expression which should always be present in the interaction of rods with arbitrarily shape objects or obstacles (cf. the Figure below). If a rod aligns its body axis perpendicular to the axis that connects the two centers of mass (situation B), overlaps are typically avoided whereas there is a significant overlap otherwise (configuration A).

Figure 2 The overlap of a rod with an object vs. the relative position.

We rewrote the corresponding paragraph to give a more mechanistic explanation, similar to the discussion above. We also hint to the new torque contribution in the context of smectic particle arrangements: this new torque introduces a coupling between local orientation order and spatial order, i.e. an smectic arrangement of particles. Furthermore, we added in the new version of the manuscript four movies, where binary collisions of rods are shown. They illustrate how the model describes the alignment of active particles with an elliptical shape.

2. As a follow-up question, I wonder if and how the new collision avoidance torque accounts for the following situation: Imagine that particle 2 collides with particle 1 ahead of particle 1's center of mass. In this case, I understand that particle 1 should rotate away from the relative position of particle 2. However, imagine that particle 2 collides behind particle 1's center of mass. In this case, I would expect particle 1 to rotate toward the relative position of particle 2. Can the new collision avoidance torque account for this effect? If not, shouldn't this effect be accounted for?

Our model accounts for the described situation: the torque, and particularly its sign, will depend on the impact parameter or, in other words, the point where rod 2 collides

into rod 1. To answer this question, we exclusively consider the symmetry of the novel torque contribution, i.e. neglecting the nematic alignment. The contribution to the torque, which rod 2 exerts on rod 1, reads

$$\dot{\varphi}_1 \propto \sin[2(\varphi_1 - \alpha_{12})],$$

where $\alpha_{12} = \arg(\mathbf{r}_1 - \mathbf{r}_2)$ is the polar angle of the vector which points from rod 2 to rod 1; the complementary angle is $\alpha_{21} = \arg(\mathbf{r}_2 - \mathbf{r}_1) = \alpha_{12} - \pi$. To simplify the discussion, we will make use of the rotational and translational invariance of our model. We place rod 1 into the origin of the coordinate system ($\mathbf{r}_1 = 0$), oriented along the x -direction ($\varphi_1 = 0$). The following sketch, where rods are represented by arrows for simplicity, illustrates all geometric quantities that are relevant for the question under consideration:

Figure 3 The geometric quantities – direction of motion and relative positions.

In this particular situation, one can simplify the torque as follows:

$$\dot{\varphi}_1 \propto \sin[2(0 - \alpha_{12})] = -\sin(2\alpha_{12}) = -\sin(2\alpha_{21}).$$

Accordingly, a torque is exerted upon rod 1 by rod 2 such that $\dot{\varphi}_1 < 0$ for $\alpha_{21} \in (0, \pi/2)$ (situation A) and positive, $\dot{\varphi}_1 > 0$, for $\alpha_{21} \in (\pi/2, \pi)$. In particular, the torque changes its sign for $\alpha_{21} = \pi/2$. Thus, the torque is different for a collision up front or in the back with respect to the orientation of rod 1. However, the discussion above neglects the presence of the nematic alignment term which shall be added in order to obtain the actual sense of rotation of rods.

The Reviewer's reasoning could be used to derive the symmetry of the new contribution to the torque. We consider a particle at position \mathbf{r}_2 in the vicinity of rod 1 with orientation φ_1 and position \mathbf{r}_1 . The repulsive interaction of both should induce a torque onto rod 1 (exerted by rod 2) that is proportional to $\mathbf{e}_{\parallel}[\varphi_1] \times (\mathbf{r}_1 - \mathbf{r}_2)/|\mathbf{r}_1 - \mathbf{r}_2|$, where the first vector will appear due to the lever arm (parallel to the rods orientation) and the second one indicates the orientation of the repulsive force (from rod 2 to rod 1). As Reviewer 1 pointed out, the sense of rotation of rod 1 depends on whether rod 2 hits rod 1 in front or in the back with respect to the center of mass of rod 1. Accordingly,

the torque must be proportional to $\mathbf{e}_{\parallel}[\varphi_1] \cdot (\mathbf{r}_2 - \mathbf{r}_1) / |\mathbf{r}_2 - \mathbf{r}_1|$, which is positive if rod 2 is in front of rod 1 with respect to its direction of motion $\mathbf{e}_{\parallel}[\varphi_1]$ and negative otherwise. The total torque $\boldsymbol{\tau}_1$ on rod 1 exerted by rod 2 is thus proportional to

$$\begin{aligned} \boldsymbol{\tau}_1 &\propto \left(\mathbf{e}_{\parallel}[\varphi_1] \cdot \frac{\mathbf{r}_2 - \mathbf{r}_1}{|\mathbf{r}_2 - \mathbf{r}_1|} \right) \left[\mathbf{e}_{\parallel}[\varphi_1] \times \frac{\mathbf{r}_1 - \mathbf{r}_2}{|\mathbf{r}_1 - \mathbf{r}_2|} \right] = \cos(\varphi_1 - \alpha_{21}) \begin{pmatrix} \cos \varphi_1 \\ \sin \varphi_1 \\ 0 \end{pmatrix} \times \begin{pmatrix} \cos \alpha_{12} \\ \sin \alpha_{12} \\ 0 \end{pmatrix} \\ &= \cos(\varphi_1 - \alpha_{21}) \begin{pmatrix} 0 \\ 0 \\ \sin(\alpha_{12} - \varphi_1) \end{pmatrix} = -\cos(\varphi_1 - \alpha_{12}) \begin{pmatrix} 0 \\ 0 \\ \sin(\alpha_{12} - \varphi_1) \end{pmatrix} \\ &= \cos(\varphi_1 - \alpha_{12}) \begin{pmatrix} 0 \\ 0 \\ \sin(\varphi_1 - \alpha_{12}) \end{pmatrix} = \frac{1}{2} \begin{pmatrix} 0 \\ 0 \\ \sin[2(\varphi_1 - \alpha_{12})] \end{pmatrix}. \end{aligned}$$

The last term is the torque contribution that follows naturally within our framework. However, there exist geometry-dependent prefactors in the actual rod model, which are not contained in the simplified discussion above; e.g., the interaction is proportional to the overlap itself – there is only a significant interaction if rods touch each other – that is not contained in the above reasoning which is based on symmetry considerations only. Hence, our framework yields all force and torque terms naturally from one principle, namely the minimization of particle overlap, such that it is not necessary to go through involved geometric consideration to construct a model for the interaction of rods.

Following the Reviewer’s suggestion, we added new animations that show binary collisions of two rods. We also show how head-on collisions can lead to polar alignment and also how rods that are aligned anti-parallel to each other, before a collision, may slide over each other, depending on the different impact parameters.

3. Given that it seems to be a generic feature of interacting rods, why wasn’t the nematic collision avoidance torque accounted for in previous studies?

This effect was contained implicitly in previous numerical studies: *de facto*, all rod-models which are based on the minimization of an overlap energy should contain this term. However, this interaction mechanism has only been studied numerically [see *Annu. Rev. Condens. Matter Phys.* **11** 441 (2020)] – for example, the first study of self-propelled rods described rods as rectangular objects [*Phys. Rev. E* **74** 030904 (2006)]. The actual strength of our study is to use a representation of rods in such a way (by Gaussian fields) that the overlap integral can be calculated explicitly. This enabled writing down the interaction energy in a simple form to derive, eventually, the contributions to the torque and, moreover, enabled performing analytical calculations based on the obtained interaction mechanism.

Breakdown of MIPS

Let us clarify the specifics of patterns arising in active matter, which are referred as MIPS. MIPS, originally observed in ensembles of self-propelled hard discs, is a phenomenon where a system spontaneously phase separates into a dense phase with local hexatic order and no orientational order, and a low density, disordered gas. In this sense, MIPS refers to a peculiar instability mechanism as well as a specific nonequilibrium phase. This phenomenon shall be contrasted from other types of density instabilities, clustering or the emergence of orientational (typically associated with density instabilities as well) order in active matter. For example, the formation of polar bands in the Vicsek model is also inherently related to the (self-propelled) motility of individual entities. However, their structural properties are fundamentally different from the specific type of MIPS.

To disentangle different types of active patterns, it is generally not sufficient to examine one order parameter. The order parameters for active systems must be multi-dimensional in order to reflect the different structural properties of active states. To be specific, the vastly different states shown in Fig. 2 of our manuscript cannot be discriminated by a single, scalar order parameter but many properties play a crucial role: local positional order, e.g hexatic or smectic, local or global orientational order that can be either of polar or nematic symmetry or the dynamics of topological defects. This is why we deliberately decided for this work to show snapshots of simulations first rather than the dependence of order parameters on motility parameters, as the interpretation of snapshots is much more direct and, therefore, enables an easier understanding of different physical regimes. Finding appropriate observables/order parameters is a nontrivial challenge.

4. On page 4, the authors explain that, even for small particle anisotropies, MIPS aggregates melt, and that hexatic order is reduced. However, besides the simulation snapshots in Fig. 2, that authors should show data supporting these claims, for example providing some measure of the presence/absence of MIPS.

We answer it in question 5.

5. How can the state of the system be described at small particle anisotropies, once MIPS clusters break down? Is there any type of order parameter, correlation, etc., that can help characterize the collective behavior of the system (besides its general disorder)?

Different ways to identify the presence/absence of MIPS were proposed, including the structure of the local density distribution (unimodal vs. bimodal), clustering identification algorithms or the calculation of macroscopic transport terms in approximate hydrodynamic theories that were used as an input for a linear stability analysis. Above, we listed several properties, which we consider to be central features of MIPS. The characterization of local density fluctuations plays, in this regard, a central role. In Fig. 6, we present a measure that quantifies density fluctuations: we calculated the variance of a coarse-grained density field and compared it to the expected variance in a Poissonian point pattern. This value is one for an homogeneous, disordered state and increases above one in the phase-separated regimes. Therefore, it can serve as a quantitative proxy

for spatial inhomogeneities. Fig. 6 reveals a rapid decrease of this measure as the aspect ratio increases above one, whereas it remains close to one for larger anisotropies.

In SI, we included additional phase diagrams in the density-activity plane for different order parameters, including the variance of local density fluctuations. Those diagrams serve as a quantitative characterization of parameter regimes where MIPS emerges and those where MIPS has broken down. Moreover, we performed a Voronoi tessellation of the system to identify neighboring particles and calculated the local hexatic order parameter. Hexatic order in high density areas is a hallmark of MIPS. Our results show that the level of hexatic order is drastically reduced as the aspect ratio is increased. Furthermore, we included additional snapshots to the SI (corresponding to the states shown in Fig. 2) where the color code indicates the level of local hexatic order. This analysis illustrates how different rod-shaped and disc-like particles behave.

Often, MIPS is identified by the bimodality/unimodality of the local density distribution. We quantified this by coarse-graining the system into boxes and subsequent counting of the local number of particles in each box. The histograms reveal a bimodal distribution for self-propelled discs in the MIPS regime, whereas it becomes unimodal as the particles become elongated. To characterize the transport of particles, we measured the mean-squared displacement vs. time for different aspect ratios (we will comment on these measures below). The respective analysis is also part of the SI.

6. The authors explain the breakdown of MIPS for slightly anisotropic particles as a result of interparticle torques. However, as they also explain, Ref. 69 recently proposed a different mechanism for very similar observations. How are these two explanations reconciled?

This is an interesting question. First, let us clarify that ref. 69 corresponds to ref. 62 of the revised manuscript. The finding that MIPS is suppressed for rod-shaped particles is similar in both works, however, the explanations are not. To our current understanding, the two explanations do not reconcile. The most obvious difference is at the level of the order parameter equations (Eq. (8) in our case).

Ref. 69 is based on "numerical evidence that the effective torque couples to the polarization but *not* to density gradients." Moreover, Ref. 69 states that the force imbalance coefficient was reduced as particles start to align locally.

In our work, we do find a coupling of the dynamics of the polar order parameter to density gradients at the macroscopic level: in Eq. (8b), there is a term (due to inter-particle torque) that is proportional to the gradient of the density with a prefactor that involves the anisotropy of rods. Our analysis shows that this term is responsible for the restabilization of the homogeneous state and, thus, the suppression of MIPS. The local alignment of particles – which Ref. 69 points out – should be a direct consequence, to our understanding, of inter-particle active torque between rod-shaped particles. However, the corresponding terms are not contained in the hydrodynamic equations in Ref. 69 – the term coupling density gradients to the dynamics of the polar order parameter is not present in Ref. 69 (cf. Appendix B of Ref. 69). This may be related

to the approximations involved in the derivation of hydrodynamic equations in Ref. 69. Our work indicates that this term is indeed relevant for understanding the formation/destabilization of aggregates via MIPS. Following the referees suggestion, we included additional comments regarding differences to Ref. 69.

7. Similarly, van Damme et al. *J. Chem. Phys.* 150, 164501 (2019) reports very similar conclusions about the suppression of MIPS by interaction torques between anisotropic particles. This paper seems to be closely related to the present manuscript. The authors should therefore also discuss the similarities and differences between their model and results and those of van Damme et al.

We thank the Reviewer for pointing out this very relevant reference to us. We included it into the revised manuscript together with a corresponding discussion of previously reported findings. van Damme et al. reports the suppression of MIPS for rod-shaped particles. This work relies mostly on comprehensive numerical simulations. Below, we list the most important differences to our work:

- van Damme et al. consider rods as spherocylinders. Their model is similar in spirit to [arXiv:1807.00294]: at first, the point of closest encounter of two rods is numerically determined and, based on the assumption of a repulsive interaction, a torque is derived by the cross product of this force and the corresponding lever arm. These geometrical quantities are generally difficult to calculate analytically. In short, the microscopic model used in our work is slightly different from the model studied by van Damme. We believe that the resulting forces and torques in our model are simpler than those in van Damme et al. It enables direct coarse-graining which we consider a central improvement in the description of self-propelled rods as it serves as the basis for all analytical considerations in our work.
- van Damme et al. assume fluctuation-dissipation relations to hold which is not the case in our study.
- van Damme et al. assume that the collective dynamics of particles can be recast in the form of an individual particle with an effective, density dependent swim speed as well as an effective rotational diffusion coefficient [Eqs. (8,9) in van Damme et al.], i.e. an effective ideal active gas. Those values were measured numerically. Our work, in contrast, starts from a microscopic model for rods and those terms (at the field theoretical level) are derived via systematic coarse-graining. It turns out that the resulting expressions are not identical.
- van Damme et al. show numerically that aggregates formed via MIPS are destabilized due to torque. They describe a mechanism for this destabilization, namely torques shorten the collision duration as rods rotate to reorient their swimming directions away from the combined center of mass. In our work, we derive those torques based on our microscopic model and show how these terms translate into nonlinear coupling at the level of order-parameter equations. In short, we also

provide a mesoscopic explanation that directly relates to the specific microscopic torque terms.

8. In general, what are the main novelties of the present manuscript with respect to Ref. 69 and van Damme et al.?

- By providing the analysis of orientational order, we go beyond the work of van Damme et al. We study, beyond the breakdown of MIPS, how the orientational order emerges.
- As outlined in our answer to question 6, we propose a microscopic mechanism that is different from the mechanism for the breakup of MIPS proposed in Ref. 69. Moreover, our work identifies which type of torque, i.e. which microscopic mechanisms at play, are responsible for the destabilization of MIPS.
- We discuss in detail how the symmetry of microscopic interactions relates to the symmetry of the coarse-grained patterns. We link the phenomena that were reported for Vicsek-type models (formation of bands) with those which were primarily studied for spatially extended, hard rods (clustering).
- We report a bistability of polar and nematic structures though the symmetry of microscopic interactions is strictly nematic. This is a central novel finding of our work. Note that in this context the work by Huber et al. [Science 2018] assumes a combination of polar *and* nematic alignment to be present at the particle level in order to explain this bistability at the macroscale, observed for microtubules.

Emergence of orientational orders

The construction of the full phase diagram of self-propelled rods is beyond the scope of this work as it is very complex and high-dimensional. Parts of these phase diagrams were studied in several manuscripts numerically, including [arXiv:1807.00294 or the paper by van Damme]. Here, we focus on some relevant cuts through the phase diagram choosing self-propulsion and aspect ratio as our control parameters. Central to our study is the combination of analytical and numerical approach to certain transitions.

9. Later on, the authors explain that local nematic order appears at small particle anisotropies for states with global disorder, and that polar order appears at higher aspect ratios, as eventually explained in Fig. 6. Again, even though the snapshots in Fig. 2 are very indicative, providing quantifications would be very useful, especially when comparing to other works.

As discussed in our answers to questions 4-8, several additional measures are now included in the SI of the revised manuscript.

10. Can the smectic order in Fig. 2e be quantified? Can we understand its emergence within the proposed theoretical framework?

To answer the question, we refer first to the answers to questions (1-3) and the corresponding figures. The emergence of smectic order is essentially related to the new

torque term discussed earlier. The coupling of relative position to orientation explains how smectic order can emerge, in contrast to simple alignment based models like in [Ginelli et al. *Phys. Rev. Lett.* **104** 184502 (2010)], where the corresponding terms are absent. In this context, the fixed points of the orientational dynamics [Eq. (3)] favor nematic alignment of particles arranged side-by-side. We highlight in this regard the complexity of the structure in Fig. 2e shows:

- a strong density instability;
- polar (orientational) ordering, both at local and global scales (at least for this system size);
- the simultaneous presence of positional and orientational order;
- convective mass transport that can also be seen from the mean-squared displacement shown in the SI of the revised manuscript;
- absence of hexatic order.

All of these features are inherently related to the activity of the system. However, all of them contrast this state from MIPS as discussed in Fig. 2a. At the current stage, it is beyond the state of the theory to predict the parameter values, where smectic ordering sets in from first principles. Nevertheless, it is important to point out that smectic order is present, in contrast to the hexatic structure that characterizes MIPS.

11. Several papers, including Refs. 33 and 36 report regimes of phase separation for self-propelled rods. Is that consistent with the findings of the present manuscript, which suggest that very small shape anisotropies readily destroy MIPS? If so, how?

[Refs. 33 and 36, correspond to refs. 34 and 37 in the revised version of the manuscript].

Phase separation of self-propelled rods is generally different from MIPS of self-propelled discs. To understand those differences, it is not enough to consider the density field only, but the local structure of states and the mesoscale dynamics have to be inspected. Furthermore, it is insufficient to examine a single order parameter. Both references mentioned by the referee do indeed point out exactly those difference of self-propelled rods and MIPS of discs and are in line with the findings of our work.

Note that the parameter regimes are clearly separated: as a function of the aspect ratio starting from disc-like particles, first MIPS breaks down, then orientational order emerges, and then rod-like particles may cluster or form macroscale superstructures.

Regarding the two references we would like to clarify the following.

Ref.: 33 – The giant aggregate, reported in Ref. 33, is not MIPS as stated in the corresponding manuscript. These aggregates are found for very long rods (aspect ratio: 4). The aggregate has a very different structure to standard MIPS aggregates – it ejects polar clusters etc.; the differences of MIPS and the giant aggregate reported in Ref.

33 are detailed in the corresponding references which one of the authors of this study co-authored, cf. the discussion of Fig. 5 in Ref. 33.

Ref.: 36 – Also in this paper, phase separation in rods is different from classical MIPS; those are different types of states. Ref. 36 calls phase-separated states *turbulent MIPS* and *glassy MIPS*, indicating that the dynamics is very different, potentially a long-time transient, and should definitely be distinguished from the states form by self-propelled discs. We expect that the routes to pattern formation, i.e. instabilities at the level of the order parameter equations, are different as well. The simplest way to see this is that the orientational order plays a crucial role for rods, which is not the case for discs. Accordingly, a description in terms of the density only cannot be applicable. As a consequence, the local structure of the rod states differs from the aggregates formed via MIPS for self-propelled discs. In this sense, different types of coarse-grained order-parameter equations should represent this system at the macroscale. Please check Fig. 6 in Ref. 36, where the authors explicitly highlight that the internal structure of the aggregates formed by rods is very different from classical MIPS. Moreover, these states appear only if there is a large anisotropy of friction coefficients (inverse mobilities) – this is a crucial difference to the parameter regime that we focused on.

12. Along the same line, are the states in Fig. 2d-e phase-separated? Should this kind of phase separation be understood differently than MIPS? Or is MIPS reentrant with aspect ratio?

Those are *different* forms of MIPS; MIPS is *not* reentrant in this respect. As we pointed out, orientational order is present in these states and their dynamics is very different from “classical” MIPS. Note that the polar order parameter is inherently related to the momentum field and, thus, to collective mass transport. The clusters formed by self-propelled rods are highly dynamic: they move actively and their persistence grows with their mass – the more particles a cluster contains, the more persistent it moves. This is in stark contrast to MIPS where the diffusion coefficient of aggregates decreases with their size. Technically speaking, MIPS is based on diffusion-mediated transport – diffusively moving drops merge and grow in size. This is very different from polar clustering of ballistically moving rods, see Ref. [Peruani & Bär *New J. Phys.* **15** 065009 (2013), Bär et al. *Annu. Rev. Condens. Matter Phys.* **11** 441 (2020)] for details. The mean-squared displacement that is now included in our work supports this argument.

Another difference is the particle dynamics at the cluster boundary. Whereas particles point inwards (Fig. 3) – MIPS aggregates are kept together by a polar boundary layer formed by particles bumping into a cluster – particles at the boundary of polar bands (Fig. 2e) are aligned with the orientation of the band. Accordingly, the mechanism which holds the band together is different from the mechanisms at play for MIPS. This implies that the “surface tension” of clusters is fundamentally different in both cases.

13. To address the two points above, it might be very insightful to show the collective diffusion coefficient Γ as a function of the aspect ratio, so that instabilities of the density field can be directly compared to instabilities of the orientational fields.

The quantification of particle transport is an important piece of information, also in the light of our answers to questions 11 and 12. We stress that the macroscale transport is not necessarily diffusive. To be more precise, the transport may be diffusive in all cases, however the crossover times to the diffusive regime are orders of magnitude larger than other relevant timescales, like the time which a particle needs to traverse the system size. Thus, the crossover is hard to observe in practise within a finite simulation time. The transport of particles in the polar band (Fig. 2e) is ballistic over significant timescales.

We included the mean-squared displacement for several parameters corresponding to the states in Fig. 2. A corresponding discussion of the different transport mechanisms helps clarifying the differences of the states shown (in addition to other facts discussed earlier, like orientational ordering). We included a corresponding paragraph to the SI and highlighted the different transport properties of those states in the main text.

Minor comments and questions

14. The text around Eq. 1 suggests that the particle anisotropy ϵ and the aspect ratio are independent parameters. Aren't they directly related?

The aspect ratio $\phi = l_{\parallel}/l_{\perp}$ and the anisotropy ϵ are indeed directly related. We rewrote the corresponding sentences to clarify this issue and expanded Eq. (1) to indicate how ϵ follows from ϕ .

15. Should one worry about the normalization of the Gaussian field that describes the particles? If Eq. S1 were normalized, the normalization factor would depend on the aspect ratio, and this dependence would carry over to the overlap, Eq. S3. So, which one is the physically correct overlap, the one with normalized ψ or the one without the normalization factor?

This is a good point, which we address briefly in the revised manuscript (Methods section).

We mention first that we do not interpret the Gaussian field ψ in the sense of a probability distribution function. Therefore, it is not appropriate to normalize its integral to one. However, it can be regarded rather as a regularization of the points in space where the rod is actually located, similar to a phase field. Therefore, ψ should carry no unit. Thus, there should be no prefactor depending on l_{\perp} or l_{\parallel} . In terms of the units, integrating this field should yield a surface, consistently.

The fields ψ were normalized in a different sense. Regarding Eqs. (M2,3): consider the overlap of two particles with the same position and the same orientation. In this case, the overlap \mathcal{I}_{kj} should reduce to the surface area of a rod in two dimensions. This is indeed the case, as one obtains $\mathcal{I}_{kj} = \pi l_{\parallel} l_{\perp}$ for $\mathbf{r}_j = \mathbf{r}_k$ and $\varphi_j = \varphi_k$. Corresponding remarks were included into the revised version of the manuscript.

16. The Methods section is referenced right before Eq. 11, suggesting that there should be a derivation of Eq. 11 in the Methods. However, I could not find the derivation in the Methods. Is it missing?

We included the basis of the kinetic theory in the Method sections **Kinetic theory** and **Coarse-grained order parameters**. The equations given in these sections, particularly Eqs. (M6) and (M7), are the basis for the derivation of Eq. (11). However, we agree that it is not obvious to see how Eq. (11) follows. Following the Reviewer's suggestion, we included additional section in the Supplementary Information where the details on this rather technical derivation are provided.

17. It would be helpful to provide details on the non-dimensionalization of the equations and the corresponding simulation units.

Following the Reviewer's suggestion, a respective discussion was included into the Method section **Numerical Langevin Simulations**.

Response to Reviewer 3

[Comments by the reviewer in black, our reply in blue]

This manuscript reports results of a detailed numerical and analytical study of anisotropic self-propelled particles with soft repulsion. The main focus of the manuscript is the investigation of the effect of elongation on the type of collective behavior – from motility induced phase separation for spheres to polar motile clusters for ellipsoids.

A theoretical framework for representing active particles by continuum fields is derived. The mechanism of destabilization of motility-induced phase separation (MIPS) and the facilitation of orientational ordering, by varying particle shape from circular to ellipsoidal, due to nonequilibrium stresses acting among self-propelled rods is elucidated. Thus, the theory bridges the gap between scalar and vectorial active matter. Depending on the parameters, like velocity and elongation, polar and nematic order may emerge and even coexist. Accordingly, the symmetry of ordered states is found to be a dynamical property of active matter.

The different collective behaviors of spherical and elongated self-propelled particles have been puzzling for quite a long time. The current study now clarifies how these differences arise, and quantifies the crossover from one to the other. Also very interesting is the coexistence of nematic and polar under certain conditions, which underlines similar observations of Ref. [42]. Therefore, I think that the manuscript is very interesting, and advances the field significantly.

We thank the Reviewer for his/her careful reading of our manuscript. We are particularly thankful that the Reviewer appreciates our work to be a significant advance of the field as well as very interesting. We took all comments and questions of the Reviewers into account, which greatly improved our manuscript. We included our answers into the main text to increase the readability of our manuscript.

The authors should consider the following comments and questions.

(1) A new model with anisotropic soft Gaussian interactions between the particles is introduced. The collective behavior of this model is then studied by particle-based simulations, in particular for small anisotropy. As far as I can see, the results are consistent with the results of previous simulations of hard discs and rods. I think this should be stated explicitly.

We thank the Reviewer for this comment and agree with his/her statement. We included respective comments in the revised manuscript along with corresponding references.

(2) Fig. 4 shows the coexistence of polar and nematic order for self-propelled rods. In which region of the parameter space can such a behavior be expected and observed?

The main control parameter for this state is the ratio of anisotropic repulsion and the self-propulsion forces. We varied the self-propulsion strength and fixed the parameters associated to the repulsive interaction potential.

For low self-propulsion speeds, we observe polar clustering as particles impede each others motion. As a consequence, positional correlations build up. This type of clustering is the same observed in [*Phys. Rev. E* **74** 030904 (2006)]. On the other hand, the high self-propulsion implies that the rods can slide past each other. In this regime, nematic alignment is found to be the dominant interaction mechanism. In the latter regime, the behavior of our system is reminiscent of Vicsek-type particles with nematic velocity alignment [*Phys. Rev. Lett.* **104** 184502 (2010)].

In between, there exists a parameter range where both states can coexist. The coexistence depends on the system size (as nematic bands do not exist in too small systems) and particle shape as well as the strength of fluctuations and the self-propulsion. The full numerical exploration of this four-dimensional parameter space is beyond the scope of this work. We therefore focused on the role of self-propulsion strength.

By systematically varying the active speed, we identified the parameter region in which this type of bistability can be expected (given that all the other parameters are fixed). In the revised manuscript, we included a panel into Fig. 4 and revised the caption such that the active speed values (i.e., strength of self-propulsion), where the bistability of polar and nematic structures is expected to emerge, are explicitly highlighted. We rewrote a comment in the main text to clarify this issue.

(3) I find Eq. (9) very interesting, as it shows that MIPS is possible also for elongated particles if the propulsion velocity is high enough. Is this consistent with simulations?

This is an interesting question, which we double checked numerically. However, Eq. (9) is a necessary, but not sufficient, condition for the emergence of MIPS. We did not find this type of restabilization. Based on our numerical results, we state that the parameter regime where MIPS is restabilized by increasing velocity is – if it exist – small. Below, we show examples of what happens if the speed is increased for slightly anisotropic particles (starting from the parameter values corresponding to the simulation in Fig. 2b). In the revision, we provide additional numerical simulations, which shows how the emergence of MIPS depends on the aspect ratio and particle speed: the SI was complemented by the phase diagrams for self-propelled discs and slightly anisotropic rods.

Figure 4 Snapshots of simulations of slightly anisotropic rods for different values of the self-propulsion force. Panel (a) is identical to Fig. 2b in the main text. Increasing the speed, a disordered state is observed (panel b). Surprisingly, nematic order emerges (the global nematic order parameter in panel c is approximate 0.82) for high self-propulsion as particles are propelled strongly enough such that they can slip past each other – as they do not impede each others motion anymore, nematic alignment is the dominant interaction mechanism, and the system does not phase separate.

We consider the following hypothetical scenarios in this context:

1. If the self-propulsion becomes comparable to the particle repulsion (cf. response to the second question), particles may slide past each other. A compact aggregate as it is formed via MIPS cannot be stable if this is the case. Note that those aggregates are typically held together by a polar boundary layer of particles which push inwards at the boundary of aggregates (Fig. 3 in the main text). If self-propulsion is too strong, rods could not push from the outside but will just slide past their interaction partners, thereby giving rise to a mixing dynamics that is different from MIPS.
2. Suppose, particles strongly repel each other and MIPS is restabilized by increasing the speed v_0 for a given (small) anisotropy. Now, one may increase the anisotropy further and destabilize MIPS again. By increasing the anisotropy, one may cross at some point the threshold of orientational (nematic or polar) ordering. This must at the latest be the case once the Onsager transition to nematic ordering is crossed, which is independent of the self-propulsion speed. As local orientational order is inherently related to mass transport and large-scale collective particle flows, MIPS aggregates will eventually break and cannot be restabilized or form at all by increasing the self-propulsion velocity v_0 . One would rather expect polar or nematic clustering to emerge in the limit of high anisotropies instead of the immobile MIPS aggregates.

The second scenario is observed in our numerical simulations: increasing the speed, density instabilities are suppressed, but a nematic phase is found for high speeds (panel c in the figure above). To underline our numerical findings that MIPS is not reentrant by increasing activity, we refer to the new phase diagrams of self-propelled discs and slightly anisotropic rods as shown in the SI in the revised manuscript. We also include a comment to the main text in the context of Eq. (9).

(4) The authors argue that the pair correlation function favors parallel (in contrast to antiparallel) alignment for large anisotropy, see Fig. 6. How can this picture be made compatible with the bistability in Fig. 4, i.e. how can nematic bands be stabilized. It seems to me that particle density must be an essential additional variable. A few remarks should be helpful.

We thank the Reviewer for pointing this out. The measurements of the pair correlation function are performed for low self-propulsion speeds and high repulsion strength, respectively. The ratio of these two effects is the main control parameter. Our arguments – that positional correlations build up and favor parallel (polar) alignment – does, strictly speaking, only hold if the self-propulsion is weaker than repulsion. Nevertheless, we illustrate the fact that polar clusters exist within the nematic band. This can be seen in the snapshot of a simulation, which is a close-up of the state represented in Fig. 4c. One may think of the nematic band as two populations of counter-propagating polar flocks (cf. a similar discussion in Ref. [*Phys. Rev. Lett.* **104** 184502 (2010)]).

Figure 5 The snapshot shows a close-up of the state represented in Fig. 4c of the main text. It illustrates that the band is comprised of polar sub-structures. The direction of motion is color-coded as indicated on the right.

Let us comment on the stability of bands, in dependence on the strength of propulsion (and repulsion strength, respectively). Let us consider the nematic band as an initial condition. For high self-propulsion velocities, particles that collide heads-on within the band may simply pass through each other, comparable to the situation in Vicsek-type models [*Phys. Rev. Lett.* **104** 184502 (2010)]. Therefore, the band can remain stable at the mesoscale. In contrast, the nematic band will be dissolved – as correctly pointed out by the Reviewer – if repulsion dominates. Now, heads-on collisions of clusters will lead to the deflection of the direction of motion, i.e. anti-parallel (nematic) alignment, cannot be sustained any longer. But particles could move in parallel next to each other. Therefore, polar alignment is the only (locally) stable configuration if repulsion among the rods is stronger than self-propulsion forces. Following the Reviewers’ suggestion, we included clarifying remarks into the revised manuscript.

We agree with the Reviewer that the density is another crucial control parameter. However, we focused here on a dilute case: the snapshots in Fig. 2 and Fig. 4 were performed at comparable small densities. We note, as commented in the main text, that at high densities, close to the percolation threshold, where particles are in constant contact with each other, the dynamics of the system is different. However, an in-depth analysis of those high-density states and their dynamics is beyond the scope of the present work.

(5) The importance of hydrodynamic flows and interactions is mentioned both in the introduction and the discussion. In fact, there is already a simulation study of the interplay of particle elongation and hydrodynamic interactions, see M. Theers et al., *Soft matter* 14, 8590 (2018). The relation of this study to the current work should be briefly discussed.

We thank the Reviewer for indicating this relevant reference to us. We included a citation along with a discussion of hydrodynamic flows and related interactions in the revised manuscript, in introduction and discussion.

REVIEWERS' COMMENTS

Reviewer #1 (Remarks to the Author):

I thank the authors for their very detailed responses and revisions. The authors have addressed all my comments and improved the manuscript. In particular, the comparison between the present and previous related work is now much better. The mechanistic intuition behind the collision avoidance torque is also very helpful. Similarly, the more precise distinction between phase separation with and without orientational order is also helpful.

I have one comment in case the authors want to consider it. In point 13 of my report, I was suggesting to look not at the particle mean-squared displacement but rather at the collective transport coefficient Γ in Eq. M9. I thought that by studying how Γ depends on aspect ratio, one could understand not only MIPS but also the phase-separation phenomena that take place for self-propelled rods. I agree with the authors that Γ alone is not enough to capture phase separation in self-propelled rods but that orientational order parameters are also required. However, since the authors already show polar and nematic transport coefficients (Eq. 11) in Fig. 6, it might be straightforward to combine both pieces of information.

Reviewer #3 (Remarks to the Author):

In their resubmittal letter, the authors have responded in detail to my previous criticism, and have addressed all points to my satisfaction. They have modified their manuscript accordingly. Therefore, I am happy to support the publication of the manuscript in Nat. Commun.

We thank again the Reviewer for careful reading of our manuscript and his/her positive judging. Here is pint-to point reply to the Reviewer comments.

Reviewer: *I have one comment in case the authors want to consider it. In point 13 of my report, I was suggesting to look not at the particle mean-squared displacement but rather at the collective transport coefficient Gamma in Eq. M9. I thought that by studying how Gamma depends on aspect ratio, one could understand not only MIPS but also the phase-separation phenomena that take place for self-propelled rods. I agree with the authors that Gamma alone is not enough to capture phase separation in self-propelled rods but that orientational order parameters are also required. However, since the authors already show polar and nematic transport coefficients (Eq. 11) in Fig. 6, it might be straightforward to combine both pieces of information.*

Reply: We thank the Reviewer for this comment. In the revised manuscript, we introduced in the main text the following paragraph (Methods section):

Beyond the identification of the transition line towards MIPS, we note that Eq. (29) could, in principle, be used to quantify numerically the transport properties of the system, as it represents the collective diffusion coefficient of the density field [61]. However, the validity of its derivation requires density fluctuations to be small (expansion to linear order in $\delta\rho$ and the dynamics of local order parameters to be fast and enslaved to the density (adiabatic elimination of $\delta\rho$). Therefore, it can only be used for homogeneous states with diffusive transport in the absence of local order and, thus, its applicability is limited in the context of self-propelled rods. That is why we show the mean-squared displacement of particles in Supplementary Note 2, a measure that quantifies the transport properties, beyond the disordered state with diffusive transport, applicable to other collective state emerging in self-propelled rods with local polar ordering.